# A helicase-tethered ORC flip enables bidirectional helicase loading

**Shalini Gupta[1], Larry J Friedman[2], Jeff Gelles[2]\*, Stephen P Bell[1]\***

[1]Howard Hughes Medical Institute, Department of Biology, Massachusetts Institute of Technology, Cambridge, United States; [2]Department of Biochemistry, Brandeis University, Waltham, United States

**Abstract** Replication origins are licensed by loading two Mcm2-7 helicases around DNA in a head-to-head conformation poised to initiate bidirectional replication. This process requires origin–recognition complex (ORC), Cdc6, and Cdt1. Although different Cdc6 and Cdt1 molecules load each helicase, whether two ORC proteins are required is unclear. Using colocalization single-molecule spectroscopy combined with single-molecule Förster resonance energy transfer (FRET), we investigated interactions between ORC and Mcm2-7 during helicase loading. In the large majority of events, we observed a single ORC molecule recruiting both Mcm2-7/Cdt1 complexes via similar interactions that end upon Cdt1 release. Between first- and second-helicase recruitment, a rapid change in interactions between ORC and the first Mcm2-7 occurs. Within seconds, ORC breaks the interactions mediating first Mcm2-7 recruitment, releases from its initial DNA-binding site, and forms a new interaction with the opposite face of the first Mcm2-7. This rearrangement requires release of the first Cdt1 and tethers ORC as it flips over the first Mcm2-7 to form an inverted Mcm2-7–ORC–DNA complex required for second-helicase recruitment. To ensure correct licensing, this complex is maintained until head-to-head interactions between the two helicases are formed. Our findings reconcile previous observations and reveal a highly coordinated series of events through which a single ORC molecule can load two oppositely oriented helicases.

**\*For correspondence:**
gelles@brandeis.edu (JG);
spbell@mit.edu (SPB)

**Competing interest:** The authors declare that no competing interests exist.

## Editor's evaluation

The initiation of DNA replication in eukaryotes is preceded by the assembly of a pre-Replicative Complex (pre-RC) at all potential origins of DNA replication during the G1 phase of the cell division cycle. The pre-RC contains a double hexamer of Mcm2-7 subunits and each hexamer eventually becomes the core of the two replicative helicases during the initiation of DNA synthesis. The current paper addresses the role of ORC and Cdc6 in loading the Cdt1-bound Mcm2-7 hexamer onto origin DNA and the data show that a single ORC molecule can load two Mcm2-7 hexamers in a sequential loading reaction that involves ORC flipping on the origin DNA. The results nicely complement other studies that show a detailed pathway for pre-RC assembly.

## Introduction

The eukaryotic DNA replication machinery first assembles and initiates synthesis at DNA sites called origins of replication. During G1, all potential origins are licensed by loading two Mcm2-7 replicative DNA helicases around the origin DNA in an inactive, head-to-head fashion (*Abid Ali et al., 2017*; *Evrin et al., 2009*; *Li et al., 2015*; *Remus et al., 2009*). This conformation prepares the helicases to initiate bidirectional replication upon activation during S phase. The Mcm2-7 complex is the core enzyme of the eukaryotic replicative helicase and its loading onto DNA is restricted to G1 phase (*Aparicio et al., 1997*; *Diffley et al., 1994*). This constraint prevents helicases from being loaded

onto replicated DNA, ensuring that no part of the genome is replicated more than once per cell cycle (*Siddiqui et al., 2013*).

Interactions between three proteins and the Mcm2-7 helicase direct eukaryotic helicase loading (*Bell and Labib, 2016*). The origin–recognition complex (ORC) binds origin DNA and then recruits Cdc6 (*Bell and Stillman, 1992*; *Speck et al., 2005*). The resulting ORC–Cdc6 complex encircles the origin DNA (*Feng et al., 2021*; *Schmidt and Bleichert, 2020*). Mcm2-7 in complex with Cdt1 associates with ORC-Cdc6 and adjacent DNA to form the short-lived ORC–Cdc6–Cdt1–Mcm2-7 (OCCM) complex (*Sun et al., 2013*; *Ticau et al., 2015*). Loading of a second Cdt1-bound-Mcm2-7 hexamer, oriented in the opposite direction of the first, completes formation of the Mcm2-7 'double hexamer'.

Multiple mechanisms have been proposed to explain how two oppositely oriented helicases are loaded at an origin. In addition to a primary ORC-binding site, natural origins include at least one additional weaker, inverted ORC-binding site (*Chang et al., 2011*; *Palzkill and Newlon, 1988*; *Wilmes and Bell, 2002*). These sequences can be located at a variety of distances from one another but are typically less than ~60 bp apart (*Chang et al., 2011*). One mechanism for helicase loading proposes that distinct ORC molecules bound at the inverted DNA sites recruit the two helicases. This model is supported by evidence that mutations in the Mcm3 C-terminus predicted to interfere with interactions between Mcm2-7 and the ORC–Cdc6 complex prevent recruitment of both the first and second helicases (*Coster and Diffley, 2017*; *Frigola et al., 2013*). In addition, for ensemble experiments two ORC DNA-binding sites are required for successful helicase loading in vitro and origin function in vivo (*Coster and Diffley, 2017*). In contrast, single-molecule helicase-loading experiments with the *ARS1* origin showed that a single ORC molecule can direct loading of both Mcm2-7 helicases (*Ticau et al., 2015*). A single ORC model is also supported by time-resolved cryoelectron microscopy (cryo-EM) experiments showing predominantly one ORC molecule bound to the DNA in each helicase-loading intermediate observed on the *ARS1* origin (*Miller et al., 2019*). A goal of the current studies is to address these apparently contradictory observations.

Structural studies have revealed important intermediates in helicase loading. ORC has been shown to bend DNA prior to Mcm2-7 recruitment (*Li et al., 2018*). Cryo-EM studies of the OCCM intermediate show that the first Mcm2-7 hexamer interacts with and encircles the DNA adjacent to ORC–Cdc6 (*Sun et al., 2013*; *Yuan et al., 2017*). These and related structures also reveal that interaction of the C-terminal region of ORC with the C-terminal region of Mcm2-7 mediates recruitment of the first Mcm2-7 hexamer (*Yuan et al., 2020*). ATP hydrolysis is required to proceed beyond the OCCM, and structures of intermediates on-pathway to the Mcm2-7 double hexamer had been elusive until a recent time-resolved cryo-EM study (*Miller et al., 2019*). Intriguingly, this study observed a complex in which an inverted ORC is engaged with the N-terminal region of the first Mcm2-7. Although it was suggested that formation of the inverted complex might be required to load double hexamers, whether one or two ORC molecules were required to form the complex and load the two Mcm2-7 helicases remained unresolved. Further, this study could not directly examine how the inverted complex was integrated into the sequence of events in helicase loading. A second goal of the current studies is to ask if a single ORC protein can mediate formation of all these stable intermediates and to understand how transitions between the intermediates are coordinated during helicase loading.

Single-molecule biochemical studies have provided kinetic and mechanistic insights into helicase loading that complement structural studies. Single-molecule studies demonstrated that the two Mcm2-7 complexes in each double-hexamer associate with origin DNA in a one-at-a-time manner (*Ticau et al., 2015*). Single-molecule FRET (sm-FRET) has been used to monitor opening and closing of the interface between Mcm2 and Mcm5 (*Bochman and Schwacha, 2008*; *Samel et al., 2014*) that provides DNA access to the central channel of the Mcm2-7 ring (*Ticau et al., 2017*). These studies revealed that Mcm2-7 is recruited with an open gate that closes around origin DNA substantially after initial DNA association. Such approaches also revealed that recruitment and loading of each Mcm2-7 involves a separate set of Cdc6 and Cdt1 molecules. Following each Mcm2-7 recruitment, Cdc6 and Cdt1 are released sequentially, and the Mcm2-7 ring closes concomitant with Cdt1 release (*Ticau et al., 2015*; *Ticau et al., 2017*). This connection between Cdt1 release and ring closing is consistent with structural data that suggest that Cdt1 holds the Mcm2-7 ring open at the Mcm2-5 gate (*Frigola et al., 2017*; *Zhai et al., 2017*).

The mechanism that drives recruitment of head-to-head Mcm2-7 hexamers remains unclear (*Bell and Labib, 2016*; *Lewis and Costa, 2020*). Here, we generate data supporting a model that reconciles

the apparently inconsistent observations regarding ORC and helicase loading and directly observe how a single ORC guides double-hexamer formation. We monitor ORC–Mcm2-7 interactions in real-time using sm-FRET and show that recruitment of each Mcm2-7 hexamer is accompanied by a short 'OM' interaction with the same ORC protein. Prior to recruiting the second Mcm2-7 hexamer, ORC forms a distinct intermediate (referred to as 'MO') with the initially loaded Mcm2-7. The transition between the OM and MO intermediates is rapid and requires Cdt1 release. Forming the MO intermediate allows ORC to release from its initial binding site and flip over the initially loaded Mcm2-7, positioning ORC to rebind DNA at an inverted binding site without release into solution. The resulting MO intermediate recruits the second Mcm2-7 and is only disrupted when Mcm2-7 double-hexamer formation is initiated. Our findings reveal a highly coordinated series of events that ensures two Mcm2-7 helicases are loaded as head-to-head pairs poised to initiate bidirectional replication.

## Results

### Monitoring ORC–Mcm2-7 interactions during helicase recruitment

To investigate the dynamics of ORC–Mcm2-7 interactions during helicase loading, we developed a sm-FRET assay for the initial interaction between these proteins based on previously described single-molecule helicase-loading experiments (*Ticau et al., 2015*). Using the structure of the OCCM as a guide (*Yuan et al., 2017*), we modified ORC and Mcm2-7 at sites that are proximal (~35 Å apart) during recruitment of the first Mcm2-7 (*Figure 1a*). ORC was labeled with a donor fluorophore at the Orc5 C-terminus (ORC$^{5C-549}$) and Mcm2-7 was labeled with an acceptor fluorophore at the Mcm2 C-terminus (Mcm2-7$^{2C-649}$). Importantly, the fluorescent labels did not interfere with protein function in ensemble helicase-loading assays (*Figure 1—figure supplement 1*). To monitor ORC–Mcm2-7 interactions during loading, purified ORC$^{5C-549}$, Mcm2-7$^{2C-649}$, Cdt1, and Cdc6 were incubated with fluorescently labeled surface-tethered origin DNA. Using total internal reflection fluorescence microscopy, we monitored the colocalization of the fluorescently modified proteins with individual DNA molecules (*Friedman and Gelles, 2015*). Alternate excitation of the donor and acceptor fluorophores (*Figure 1—figure supplement 2b*) allowed observation of the association of each labeled protein with origin DNA, and determination of the apparent FRET efficiency ($E_{FRET}$) during donor excitation measured ORC–Mcm2-7 C-terminal interactions (*Figure 1a, b*). We will refer to the ORC–Mcm2-7 interactions monitored using FRET between ORC$^{5C-549}$ and Mcm2-7$^{2C-649}$ as 'OM interactions'.

We focused our studies on SM event records that are consistent with Mcm2-7 double-hexamer formation. Mcm2-7 complexes associated with DNA in a one-at-a-time fashion as described previously (*Ticau et al., 2015*). A fraction of the DNAs (~28% of total DNAs) showed two stepwise increases in Mcm2-7$^{2C-649}$ fluorescence intensity as expected for formation of the Mcm2-7 double hexamer. Of the DNAs with two Mcm2-7 associations, we limited further analysis to a subset showing two Mcm2-7 complexes that are retained on DNA for 20 or more frames of acquisition (>48 s). These long-lived sequential associations occurred on 12% of DNAs and represent successful Mcm2-7 double-hexamer formation (*Ticau et al., 2015*). The remaining sequential associations (on 16% of DNAs) were short-lived (<48 s), and we considered these to be unsuccessful instances of helicase loading (*Ticau et al., 2015*).

Monitoring ORC and Mcm2-7 DNA association showed that for most double-hexamer formation events, one ORC molecule loaded two Mcm2-7 complexes. Specifically, we observed association of one ORC molecule with the DNA throughout the interval for sequential recruitment of two Mcm2-7 molecules in 81% of double-hexamer formation events. The observation of only one ORC is not due to incomplete ORC labeling as we determined that 88 ± 2% of the ORC$^{5C-549}$ protein was labeled (see Materials and methods). Thus, if a second ORC was required for recruiting the two helicases, we would expect to see association of two ORCs in 77% ($0.88^2 = 0.77$) of double-hexamer formation events. In contrast, we observed two ORC proteins during sequential Mcm2-7 recruitment in only 13% of the double-hexamer formation events (22/166). The remaining 6% of double-hexamer formation events had three associated ORC molecules, likely reflecting one or more nonspecific ORC DNA-binding events.

In the small fraction of double-hexamer formation events with two ORC molecules, the arrival of the second ORC was not coordinated with any other event we could observe during helicase loading. We found two types of two-ORC double-hexamer formation events. In 8/166 events, different ORC

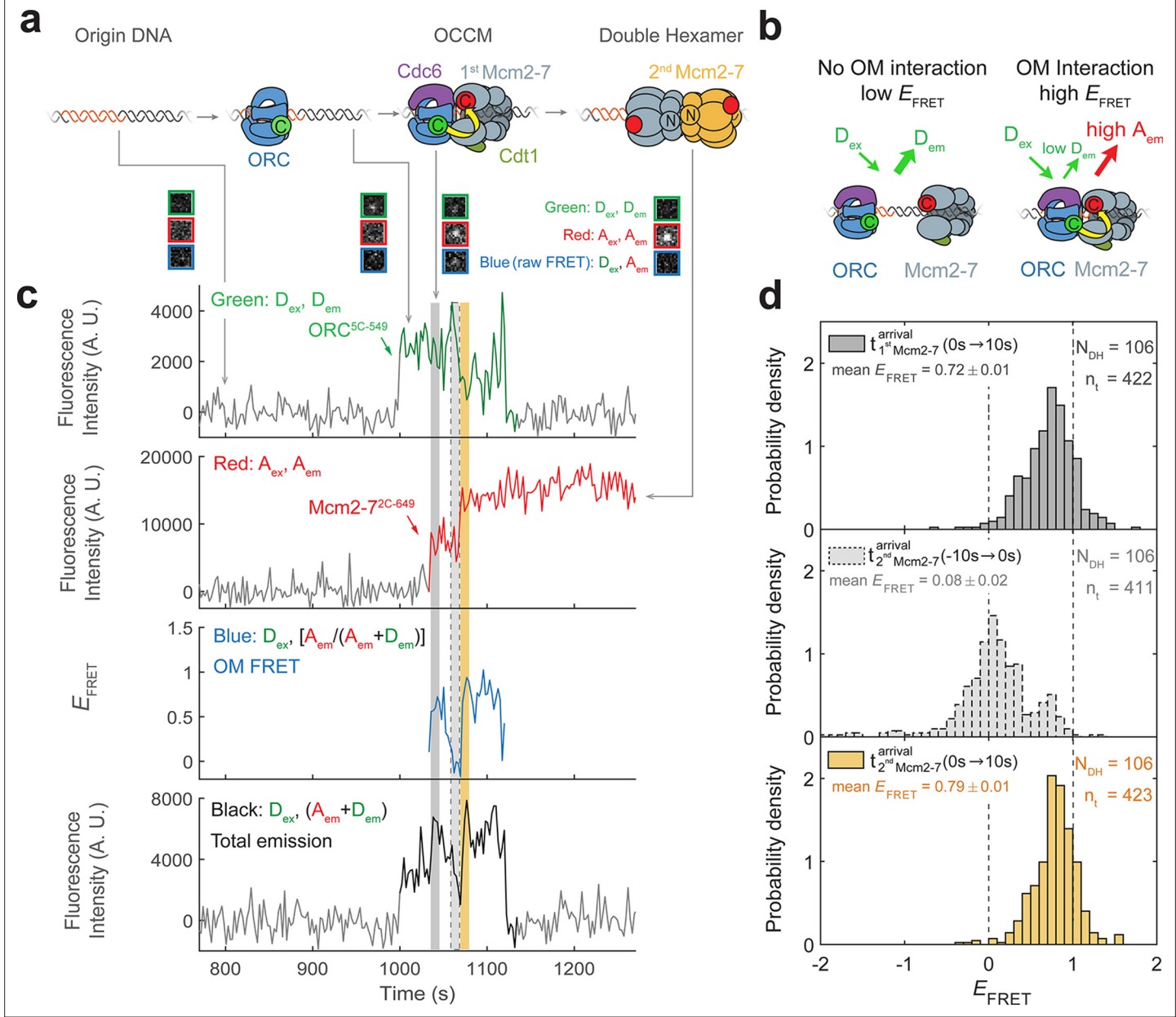

**Figure 1.** Origin–recognition complex (ORC) recruits two Mcm2-7 helicases through sequential OM interactions. (**a**) A schematic of the molecular events during helicase loading. The ORC and Mcm2-7 proteins are labeled at their C-termini (C) which form an interface during first Mcm2-7 recruitment (*Yuan et al., 2017*). ORC[5C-549] is labeled with a donor fluorophore (D; green circles) and Mcm2-7[2C-649] is labeled with an acceptor fluorophore (A; red circles). The associated fluorescence images show single frames of ORC, Mcm2-7, and raw FRET fluorescence spots at a single DNA molecule in green, red, and blue outlines, respectively. We use gray and yellow coloring to indicate the first and second Mcm2-7 hexamers, respectively. (**b**) Schematic of DNA-bound ORC[5C-549] and Mcm2-7[2C-649] proteins. When ORC[5C-549] and Mcm2-7[2C-649] are not associated, excitation of the donor ($D_{ex}$) only yields emission from the donor fluorophore ($D_{em}$). However, when ORC[5C-549] and Mcm2-7[2C-649] are in proximity, we observe emission from the acceptor ($A_{em}$) on donor excitation due to FRET, and corresponding lower emission from the donor fluorophore ($D_{em}$). (**c**) A representative trace showing ORC and Mcm2-7 associations with DNA and ORC–Mcm2-7 (OM) interactions during helicase loading. Donor-excited fluorescence record shows ORC[5C-549] association (green: $D_{ex}$, $D_{em}$ panel) and acceptor-excited fluorescence record shows Mcm2-7[2C-649] association (red: $A_{ex}$, $A_{em}$ panel). Gray arrows link to the corresponding molecular events shown in (**a**). Interaction between ORC and Mcm2-7 decreases the distance between the donor and acceptor fluorophores and results in increased apparent FRET efficiency ($E_{FRET}$, blue: $D_{ex}$, [$A_{em}$/ ($A_{em}$ + $D_{em}$)] panel). $E_{FRET}$ values are calculated using donor-excited emission from the donor and acceptor fluorophores (see Materials and methods) and they are therefore shown only at times when both a donor and acceptor fluorophore are present on the DNA molecule. The black: $D_{ex}$, ($A_{em}$ + $D_{em}$) panel shows donor-excited total emission (see *Figure 1—figure supplement 3*). An objective image-analysis algorithm (*Friedman and Gelles, 2015*) detects a spot of fluorescence at time points shown in green, red, and black on the time records. Gray, gray-dash, and yellow highlight three 10-s time intervals referenced in (**d**). A.U., arbitrary units. Concentrations

*Figure 1 continued on next page*

*Figure 1 continued*

of labeled proteins in the reaction are 0.5 nM ORC$^{5C-549}$, 15 nM Cdt1–Mcm2-7$^{2C-649}$. **Figure 1—figure supplement 2a** shows additional records. (**d**) Histogram plots of $E_{FRET}$ values for 106 single-ORC-mediated, double-hexamer formation events during three 10-s time intervals: immediately after the first (top) or second Mcm2-7 (bottom) arrives or before the second Mcm2-7 arrives (middle). Examples of these intervals are indicated in (**c**) with gray, yellow, and gray-dash, respectively. The two dashed lines indicate $E_{FRET}$ values of 0 and 1. $N_{DH}$, number of double-hexamer formation events and $n_t$, number of signal points. Rare $E_{FRET}$ values below −2 and above +2 were excluded (2/428, 13/428, and 1/428 signal points from the top, middle, and bottom histograms, respectively).

The online version of this article includes the following source data and figure supplement(s) for figure 1:

**Source data 1.** Primary data for histograms in **Figure 1d**.

**Figure supplement 1.** Fluorescently labeled proteins function in ensemble helicase-loading assays.

**Figure supplement 1—source data 1.** Primary data and quantification for graph in **Figure 1—figure supplement 1**.

**Figure supplement 2.** Each Mcm2-7 arrival is associated with an OM interaction.

**Figure supplement 3.** Enhancement of total emission upon donor excitation during OM interactions is due to protein-induced fluorescence enhancement (PIFE) exhibited by ORC$^{5C-549}$.

**Figure supplement 4.** The end of the second OM interaction is concomitant with origin–recognition complex (ORC) departure in a majority of helicase-loading events.

molecules were present during the two Mcm2-7 associations. The first ORC appeared to release from DNA following the recruitment of the first Mcm2-7, and a second ORC was present during the recruitment of the second Mcm2-7. These rare events provide the first direct evidence of helicase loading mediated by two separate ORC molecules. For 14/166 events, one ORC molecule was present throughout and an additional, second ORC associated between the first and second Mcm2-7 recruitment events. We note that for this category, it is possible that a single ORC molecule mediates both recruitment events with the second ORC being DNA bound but uninvolved (see 'The same ORC molecule interacts with both Mcm2-7 helicases').

We restricted subsequent analysis to the one ORC double-hexamer formation events for two reasons. First, they were the most frequent events observed. Second, we were particularly interested in investigating how one ORC could load two Mcm2-7 helicases in opposite orientations.

FRET between ORC$^{5C-549}$ and Mcm2-7$^{2C-649}$ is a reporter for C-terminal ORC–Mcm2-7 interactions during helicase recruitment (**Figure 1b**). When both ORC$^{5C-549}$ and Mcm2-7$^{2C-649}$ were colocalized with DNA, we observed transient periods of high $E_{FRET}$ upon Mcm2-7 DNA association (**Figure 1c**, blue: $D_{ex}$, [$A_{em}/(A_{em} + D_{em})$] panel). Mean $E_{FRET}$ ± standard error of the mean (SEM) in the 10-s intervals after first and second Mcm2-7 arrivals were 0.72 ± 0.01 and 0.79 ± 0.01, respectively (**Figure 1d**, top and bottom panels). Three findings give us confidence that these high $E_{FRET}$ values arise from interactions between ORC and Mcm2-7. First, we positioned the fluorescent probes on ORC and Mcm2-7 to detect interactions between regions that are proximal in the complex presumed to represent the initially recruited Mcm2-7, the OCCM (**Yuan et al., 2017**). Second, in 105/111 instances where a first Mcm2-7$^{2C-650}$ arrived at an ORC$^{5C-549}$-bound DNA, high $E_{FRET}$ was detected at the same time as the arrival of the Mcm2-7 on DNA (temporal resolution is ±2.4 s; **Figure 1—figure supplement 2b, c**). This coordination indicates that the high $E_{FRET}$ value monitors an interaction occurring during initial Mcm2-7 recruitment. Finally, we verified that observing high $E_{FRET}$ was dependent on the placement of the fluorescent probes on Mcm2-7 and ORC. Moving the label on either ORC or Mcm2-7 to the opposite side of the protein relative to the C-terminal ORC–Mcm2-7 interface seen in the OCCM resulted in strong reductions in the associated $E_{FRET}$ values (**Figure 2—figure supplement 1**, **Figure 3—figure supplement 1**). Together, these observations establish that the high $E_{FRET}$ state arises from specific ORC–Mcm2-7 interactions during initial recruitment of Mcm2-7.

We note that the total emission upon donor excitation is enhanced during OM interactions (**Figure 1c**, black: $D_{ex}$, ($D_{em} + A_{em}$) panel). This is due to protein-induced fluorescence enhancement (**Hwang et al., 2011**) of the of the DY549P1 fluorophore during the OM interaction (**Figure 1—figure supplement 3**). Importantly, this effect does not hinder our ability to detect OM interactions. Instead, it represents an additional indicator of this interaction.

## Each Mcm2-7 recruitment is accompanied by an OM interaction

Monitoring the ORC–Mcm2-7 recruitment interface revealed two periods of OM interaction during helicase loading. Arrival of each of the two Mcm2-7 helicases was accompanied by a short period of high $E_{FRET}$ in almost all cases (*Figure 1c*, blue: $D_{ex}$, [$A_{em}/(A_{em}+D_{em})$] panel). In 111 examples of double-hexamer formation, 106 showed distinct periods of high $E_{FRET}$ with each of the two Mcm2-7 arrivals (see below for a description of the remaining 5/111 cases). The elevated $E_{FRET}$ values were simultaneous with the arrival of the corresponding Mcm2-7 on the DNA (*Figure 1—figure supplement 2c*), consistent with both OM interactions mediating Mcm2-7 recruitment. The start time of the first OM interaction matched the first Mcm2-7 arrival in 104/106 cases and the start of the second OM interaction matched the arrival of the second Mcm2-7 in 98/106 cases (time resolution of experiment ~ 2.4 s). The rare cases (2/106 and 7/106 for the first and second Mcm2-7, respectively) that were not simultaneous were all within 4.8 s of the corresponding Mcm2-7 arrival except for one (6 s). The similar mean $E_{FRET}$ values associated with the first and second OM interactions (0.72 ± 0.01 and 0.79 ± 0.01, respectively; *Figure 1d*) indicate that recruitment of the first and second Mcm2-7 is mediated by the formation of a similar ORC–Mcm2-7 interface.

The first OM interaction consistently ended before the second Mcm2-7 was recruited during helicase loading. Although both ORC and first Mcm2-7 remained associated with the DNA, the OM-FRET signal was lost before second Mcm2-7 arrival in 106/106 cases. Thus, there was always a period lacking OM interaction between the two Mcm2-7 recruitment events. Consistent with this observation, $E_{FRET}$ values in the 10-s interval preceding the second Mcm2-7 arrival were low (*Figure 1d*, middle panel), with an average value of 0.08 ± 0.02. The second OM interaction observed was also short lived and in a majority of events ended with the loss of ORC$^{5C-549}$ fluorescence (71/106; *Figure 1—figure supplement 4a*). We confirmed this loss of donor fluorescence was typically not due to photobleaching (*Figure 1—figure supplement 4b*), indicating that the end of the second OM interaction in these cases was concomitant with the departure of ORC from the DNA. These data indicate that the interactions that recruit the first Mcm2-7 are always broken before the second Mcm2-7 is recruited, as expected if the same ORC mediated both events.

## ORC interacts with both the first and second Mcm2-7 during double-hexamer formation

Two distinct mechanisms of helicase loading could result in the observation of two sequential OM interactions with a single ORC protein. The two high $E_{FRET}$ signals could arise due to ORC$^{5C-549}$ interacting twice with the first Mcm2-7$^{2C-649}$ (*Figure 2a*, Model 1, 'Re-FRET'). Alternatively, ORC$^{5C-549}$ may instead interact with each of the two Mcm2-7$^{2C-649}$ complexes on arrival at the DNA (Model 2). To achieve the head-to-head conformation of the Mcm2-7 helicases in the double hexamer, Model 2 would require either ORC or the first Mcm2-7 to invert its orientation on the DNA. Since initial loading of the Mcm2-7 complex involves passing origin DNA through the Mcm2-5 gate followed by gate closing, inversion of Mcm2-7 on the DNA would require reopening of the Mcm2-5 gate. However, previous studies found no evidence of Mcm2-5 gate reopening during helicase loading (*Ticau et al., 2017*). For this reason, the version of Model 2 in which ORC rebinds the DNA on the opposite side of the first Mcm2-7 and in the opposite orientation is most likely. We call this the 'ORC-flip' model.

To distinguish between the Re-FRET and ORC-flip models, we performed the SM helicase-loading reaction with a mixture of two Mcm2-7 preparations labeled at different positions with acceptor fluorophores (Mcm2-7$^{2C-649}$ and Mcm2-7$^{4N-650}$). Importantly, in contrast to Mcm2-7$^{2C-649}$, the N-terminally labeled Mcm2-7$^{4N-650}$ exhibits very low FRET with ORC$^{5C-549}$ (*Figure 2—figure supplement 1*). Mixing an equimolar ratio of these two modified Mcm2-7 complexes with ORC$^{5C-549}$ allows the formation of four distinct populations of double hexamers (*Figure 2b*). We will refer to the four populations of double hexamers by the position of the acceptor fluorophore on the first and second helicases in that order. Thus, a double hexamer formed where the first helicase is Mcm2-7$^{2C-649}$ and second is Mcm2-7$^{4N-650}$ will be referred to as 'CN' and a double hexamer where the first helicase is Mcm2-7$^{4N-650}$ and second is Mcm2-7$^{2C-649}$ is 'NC'. Double hexamers with the same labeled Mcm2-7 for the first and second events would be 'CC' or 'NN.'.

Depending on which mechanism is used, the four double-hexamer populations will generate distinct OM interaction profiles (*Figure 2b*). If the Re-FRET model is accurate, ORC would only interact with the first recruited helicase. Since the CC and CN double hexamers have Mcm2-7$^{2C-649}$

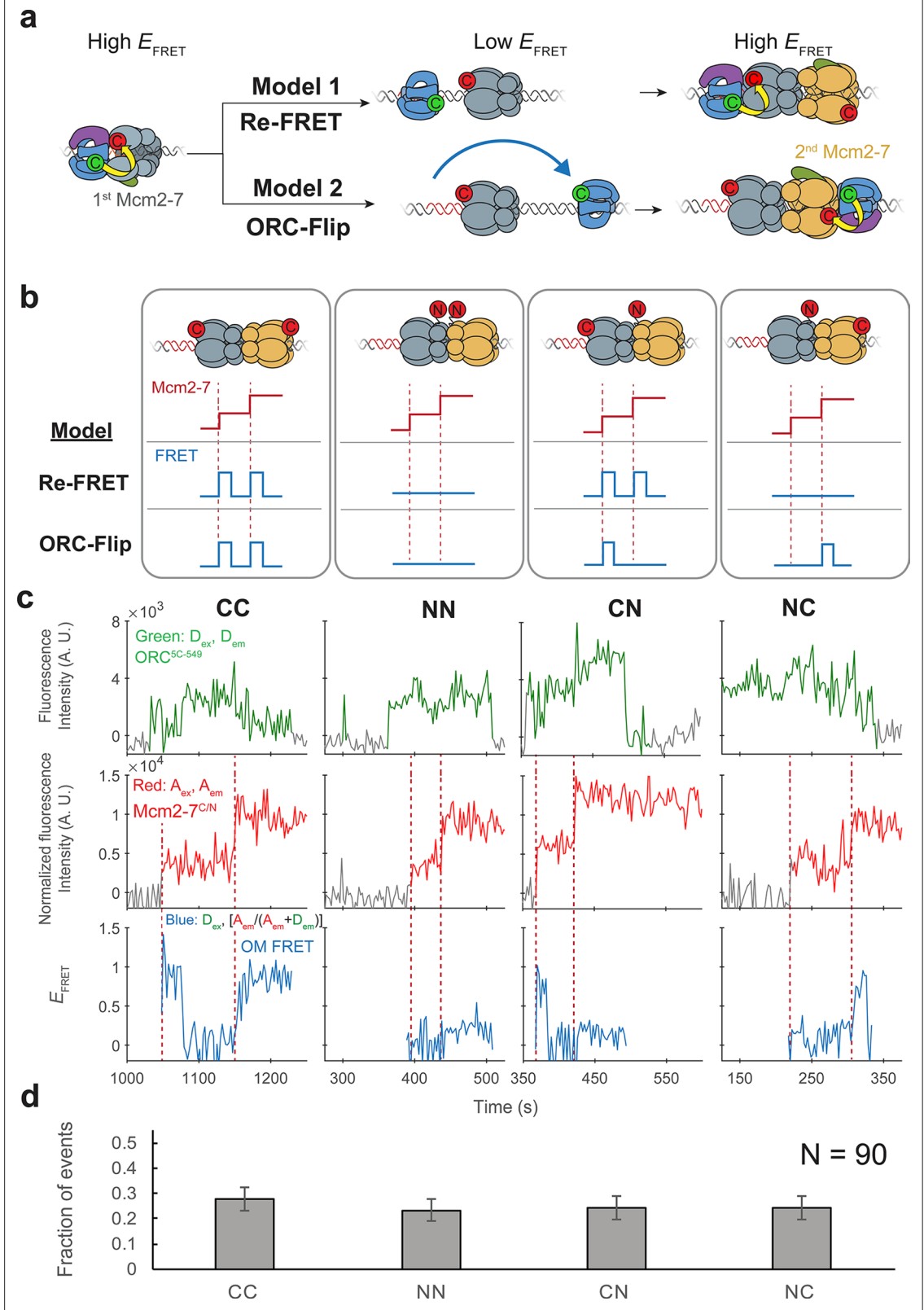

**Figure 2.** ORC makes OM interactions sequentially with the first and second Mcm2-7 helicases. (**a**) Two models to explain an OM-FRET interaction accompanying each Mcm2-7 arrival in an experiment (*Figure 1*) in which both ORC and Mcm2-7 are C-terminally labeled. Both models begin with ORC recruiting the first Mcm2-7 via the first OM interaction (high $E_{FRET}$). In the Re-FRET model, ORC separates from the first Mcm2-7 (resulting in low $E_{FRET}$) followed by reinteracting with the first Mcm2-7. In the ORC-flip model, ORC releases from its original binding site and rebinds the DNA at a

*Figure 2 continued on next page*

*Figure 2 continued*

second inverted binding site on the other side of Mcm2-7, resulting in lower $E_{FRET}$. The flipped ORC then recruits the second Mcm2-7 via a second OM interaction. (**b**) Experimental setup to distinguish between the Re-FRET and ORC-Flip models. Acceptor-labeled Mcm2-7$^{2C-649}$ (C; red circles) and Mcm2-7$^{4N-650}$ (N; red circles) are mixed in an equimolar ratio, resulting in four subpopulations of double hexamers – CC, NN, CN, and NC. Red dashed lines indicate Mcm2-7 arrival times during double-hexamer formation. Only the C-terminally labeled Mcm2-7$^{2C-649}$ molecules exhibit high FRET with donor-labeled ORC$^{5C-549}$. The Re-FRET and ORC-flip models predict distinct FRET profiles for the four double-hexamer populations generated. Importantly, the observation of single FRET peaks is unique to the ORC-flip model. (**c**) Representative double-hexamer formation events from an experiment with mixed Mcm2-7$^{2C-649}$ and Mcm2-7$^{4N-650}$. Concentrations of labeled proteins in the reaction are 0.5 nM ORC$^{5C-549}$, 7.5 nM Cdt1–Mcm2-7$^{2C-649}$ and 7.5 nM Cdt1–Mcm2-7$^{4N-650}$. *Figure 2—figure supplement 2* shows additional records of CN and NC double hexamers. (**d**) The fraction of observed events (± standard error, SE) corresponding to the type of FRET profile shown above in (**c**).

The online version of this article includes the following source data and figure supplement(s) for figure 2:

**Source data 1.** Primary data for graph in *Figure 2d*.

**Figure supplement 1.** Mcm2-7$^{4N-650}$ does not exhibit high $E_{FRET}$ with ORC$^{5C-549}$.

**Figure supplement 2.** Additional fluorescence intensity traces for the mixed double-hexamer populations with single OM-FRET peaks observed in *Figure 2c*.

---

as the first helicase, these populations should have two sequential high FRET peaks corresponding to each helicase arrival. In contrast, because the NN and NC double hexamers have Mcm2-7$^{4N-650}$ as the first helicase, the Re-FRET model predicts these populations should not exhibit high FRET during double-hexamer formation.

If the ORC-flip model is correct, ORC would form an OM interaction sequentially with each Mcm2-7 helicase in the double hexamer. Thus, for this model only CC double hexamers should be associated with two sequential high FRET peaks. Importantly, for the ORC-flip model the mixed double hexamers (CN and NC) would exhibit a single high FRET peak when the C-terminally labeled Mcm2-7$^{2C-649}$ arrives. Thus, the observation of single OM interaction $E_{FRET}$ (OM-FRET) peaks during loading of two helicases is unique to the ORC-flip model, allowing us to distinguish between the two models.

We observed four distinct patterns of OM interaction profiles at similar frequencies when the differently labeled Mcm2-7 complexes were present (*Figure 2c*, *Figure 2—figure supplement 2*). Of 90 double-hexamer formation events, we identified 25 events with 2 OM-FRET peaks and 21 events with no associated OM-FRET. Strikingly, approximately half the double-hexamer formation events had a single OM-FRET peak associated with them (44/90), consistent with the ORC-flip model (see *Figure 2b*, CN and NC panels). Also consistent with this model, these single OM-FRET events are equally distributed between events in which FRET is associated with the first (22/90 events) or the second (22/90 events) Mcm2-7 arrival (*Figure 2d*). Indeed, based on the $E_{FRET}$ patterns, we were able to infer the underlying double-hexamer 'type' for each OM-FRET pattern revealing similar frequencies of CC, NN, CN, and NC double hexamers. Both the presence of the four patterns of FRET and their equal frequency strongly support the ORC-flip model. We note that these patterns cannot be explained by incomplete labeling of Mcm2-7 complexes, as we only analyzed events with two labeled Mcm2-7 and a labeled ORC.

In the experiment with only C-terminally labeled Mcm2-7 (*Figure 1*), most events exhibited two sequential OM-FRET peaks, but rare events did not. We observed 1/111 events in which only the first Mcm2-7 arrival exhibited FRET and 3/111 events in which only the second Mcm2-7 arrival exhibited FRET. We also observed 1/111 events with no FRET on either Mcm2-7 arrival. Although all five events appear to have a single ORC colocalized with DNA, these events are likely to arise from DNAs with two bound ORC molecules, one of which is unlabeled. Importantly, these events occur at very low frequency and cannot explain the frequent observation of one or no FRET peaks observed in reactions with mixed Mcm2-7$^{2C-649}$ and Mcm2-7$^{4N-650}$ complexes (*Figure 2d*). Thus, we conclude that ORC makes OM interactions with both the first and second Mcm2-7 helicases during helicase loading.

## The same ORC molecule interacts with both Mcm2-7 helicases

Although we saw fluorescent signals consistent with a single ORC colocalized with DNA during helicase loading, it remained possible that a rapid exchange of ORC molecules occurred between the first and second Mcm2-7 recruitments. Two models (*Figure 3a*) could explain how ORC might interact with both Mcm2-7 helicases on recruitment in a manner that is consistent with our previous results.

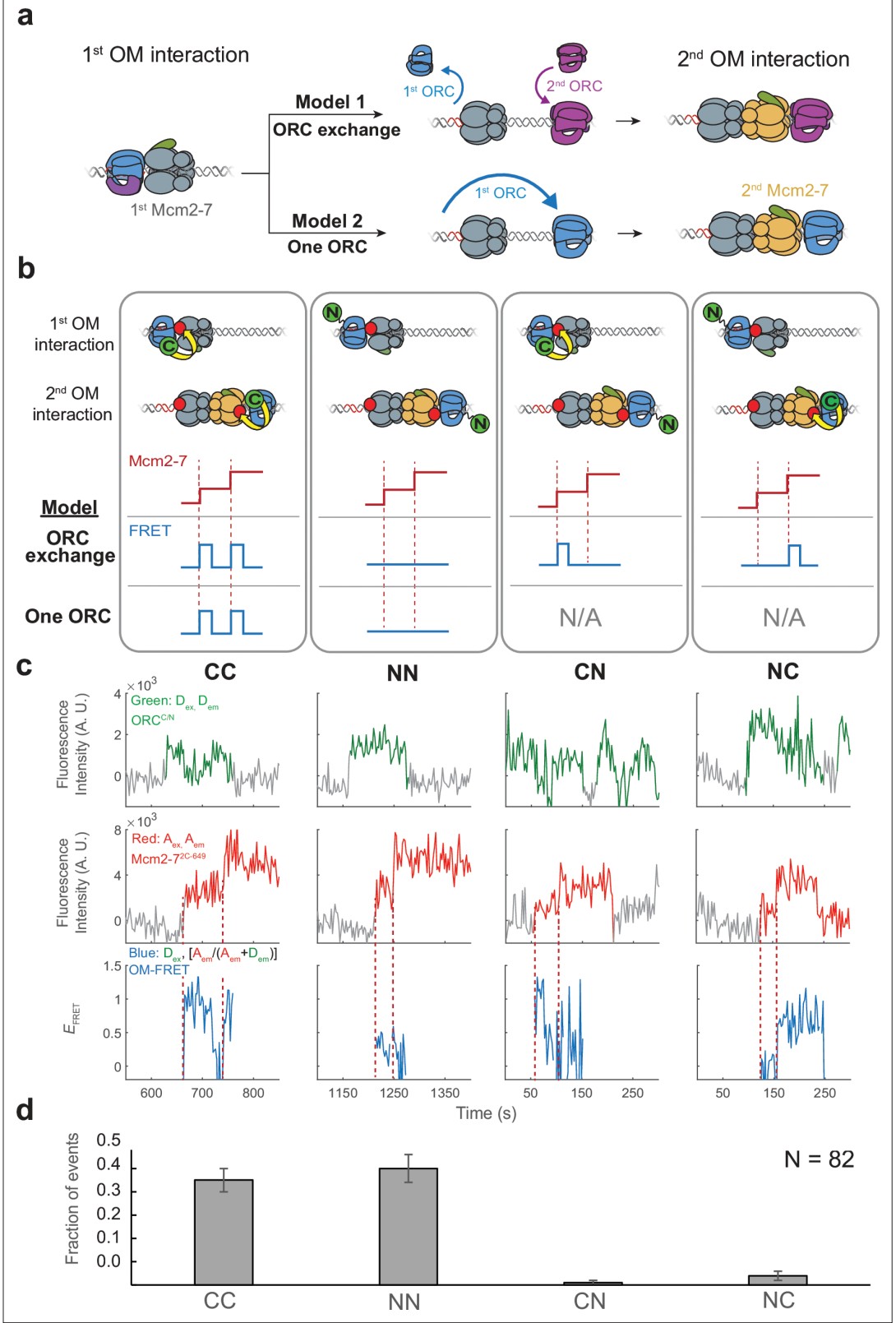

**Figure 3.** The same ORC molecule interacts with both Mcm2-7 helicases. (**a**) Two models that can explain how ORC interacts with both Mcm2-7 helicases. Both models are consistent with the observation of uninterrupted fluorescence of a single ORC throughout both helicase recruitment events. In the 'ORC-exchange' model, what appears to be a single, continuously present ORC is in fact rapid and coordinated exchange of two ORC molecules that recruit the first and second Mcm2-7 helicases, respectively. In the 'one-ORC' model, the same ORC recruits both helicases. (**b**) Experimental setup to

*Figure 3 continued on next page*

*Figure 3 continued*

distinguish between the ORC-exchange and one-ORC models. A mixture of donor-labeled ORC$^{5C-549}$ (C; green circles) and ORC$^{1N-550}$ (N; green circles) is added to a reaction with acceptor-labeled Mcm2-7$^{2C-649}$. Only ORC$^{5C-549}$ (C-terminally labeled) can FRET with Mcm2-7$^{2C-649}$ (*Figure 3—figure supplement 1*). Red dashed lines indicate Mcm2-7 arrival times. The predicted FRET results are illustrated for each model. There are four possible types of ORC pairs and corresponding FRET results when two ORC molecules mediate first and second OM interactions – CC, NN, CN, and NC. However, in the one-ORC model only the CC and NN pairs are possible resulting in only two possible FRET results. N/A, not applicable. (**c**) Representative double-hexamer formation events in an experiment with mixed ORC$^{5C-549}$ and ORC$^{1N-550}$. Concentrations of labeled proteins in the reaction are 0.25 nM ORC$^{5C-549}$, 0.25 nM ORC$^{1N-550}$, and 15 nM Cdt1-Mcm2-7$^{2C-649}$. (**d**) The fraction of events that correspond to each type of FRET profile shown above in (**c**).

The online version of this article includes the following source data and figure supplement(s) for figure 3:

**Source data 1.** Primary data for graph in *Figure 3d*.

**Figure supplement 1.** ORC$^{1N-550}$ does not exhibit high $E_{FRET}$ with Mcm2-7$^{2C-649}$.

First, two ORCs could be involved in helicase loading, with a separate ORC molecule guiding the recruitment of each of the two Mcm2-7s. To be consistent with our previous observations, the two ORCs would need to exchange in a rapid and coordinated manner such that only one ORC appears colocalized with DNA at any time. We will refer to this as the 'ORC-exchange' model. An alternative model posits that the same ORC interacts with both the first and second helicases ('one-ORC' model). In this case, the same ORC would release from its initial DNA-binding site and rebind the DNA in the opposite orientation prior to recruitment of the second Mcm2-7.

To distinguish between the ORC-exchange and one-ORC models, we used a mixed-ORC-labeling approach (*Figure 3b*). Analogous to the mixed-Mcm2-7-labeling used in *Figure 2*, we performed the helicase-loading reaction with Mcm2-7 labeled in a single position (Mcm2-7$^{2C-649}$) and a mixture of ORC proteins labeled at distinct sites (ORC$^{5C-549}$ and ORC$^{1N-550}$). Only the C-terminally labeled ORC protein shows significant FRET with Mcm2-7$^{2C-649}$ (*Figure 3—figure supplement 1*). In the ORC-exchange model, four possible pairs of ORC molecules could mediate helicase loading. We will refer to these ORC pairs as CC, NN, CN, or NC based on the position of the fluorophore on the first and second ORC molecules, respectively. For example, the mixed ORC pair CN indicates ORC$^{5C-549}$ as the first ORC and ORC$^{1N-550}$ as the second. In contrast, the one-ORC model requires that a single ORC mediate both OM interactions. Thus, for this model, only two of the four possible ORC 'pairs' described above (CC and NN) are possible because the same ORC recruits the first and second Mcm2-7.

We observed two predominant patterns of OM interaction profiles in the experiment with the mixed ORC proteins (*Figure 3c*). Of 82 double-hexamer formation events, 37 events had two OM-FRET peaks (fraction of events is 0.45 ± 0.05; *Figure 3d*) and 41 events had no associated OM-FRET (0.50 ± 0.06). The profiles with two OM-FRET peaks can only come from CC ORC 'pairs' and those with no OM-FRET are characteristic of NN ORC 'pairs' (see CC and NN panels in *Figure 3b*). We identified 1/82 events that was consistent with a CN ORC pair (0.01 ± 0.01) and 3/82 events with NC ORC pairs (0.04 ± 0.02). The ORC-exchange model predicts that the fraction of events with CC and NN ORC pairs should be similar to those for the mixed ORC pairs (CN and NC). However, the vast majority of events are of CC and NN types (78/82), which is consistent with the same ORC interacting with both Mcm2-7 helicases. The rare CN and NC events are most likely two ORC events in which one ORC is unlabeled (see above). Based on these findings, we conclude that the majority of events involve one ORC molecule interacting sequentially with both helicases (one-ORC model).

The above analyses included only events in which a single fluorescent ORC was present throughout loading. We also analyzed 27 events in which more than one fluorescent ORC was present during the loading event. In a subset of these events (7/27) the first ORC disappeared before the second one arrived, suggesting a two-ORC mechanism. We also observed 20/27 events in which a second ORC associated between the first and second Mcm2-7 recruitment events while the first ORC remained present throughout. Interestingly, almost all (18/20) of those events showed either the CC or the NN pattern consistent with the first ORC mediating both recruiting events. These observations suggest that the fraction of helicase-loading events mediated by two ORCs is lower than the small fraction in which two ORCs were present.

## Cdt1 release corresponds with the end of OM interaction

Measuring the duration of OM interactions revealed a kinetic difference between the first and second OM interactions. In each case, the OM interactions are relatively short-lived, however, the lifetime distributions of first and second events are distinct. A survival plot of the two OM interactions (*Figure 4a*) shows the distribution of OM interaction time during first Mcm2-7 recruitment is significantly shorter (mean ± SE, 30 ± 2 s) than that during recruitment of the second Mcm2-7 helicase (mean, 56 ± 4 s). Interestingly, the average OM interaction durations are similar to the mean dwell times of the first and second Cdt1 molecules reported previously (*Ticau et al., 2015*). This similarity raised the possibility that the release of Cdt1 that occurs during loading of each Mcm2-7 is connected to the termination of the OM interaction.

To examine the possibility that Cdt1 release is linked to the end of the corresponding OM interaction, we monitored Cdt1 release and OM-FRET in a single experiment. To this end, in addition to the previously described fluorescently labeled OM-FRET pair, we included Cdt1 labeled at its N-terminus with a red-excited acceptor fluorophore (Cdt1$^{N-649}$). Mcm2-7 and Cdt1 arrive together at origin DNA (*Ticau et al., 2015*). Thus, in this experiment each Mcm2-7/Cdt1 arrival results in the association of two red fluorophores with DNA (*Figure 4b*, red: $A_{ex}$, $A_{em}$ panel). Successful helicase loading is associated with the release of Cdt1 while Mcm2-7 remains bound to the DNA (*Ticau et al., 2015*). Such Cdt1-release events will result in the loss of one red fluorophore and appear as the reduction of the acceptor fluorescence intensity (red: $A_{ex}$, $A_{em}$). Although Cdt1$^{N-649}$ is also labeled with an acceptor fluorophore, this site of labeling does not exhibit strong FRET with ORC$^{5C-549}$ (*Figure 4—figure supplement 1*). In a control experiment in which only Cdt1 is labeled with the acceptor fluorophore, the average $E_{FRET}$ observed when Cdt1 is present was 0.358 ± 0.003, which is easily distinguished from that of the OM-FRET pair (average $E_{FRET}$ 0.72 ± 0.01; see *Figure 1d*). However, the presence of Cdt1$^{N-649}$ does contribute to a slightly elevated $E_{FRET}$ when the OM interaction is formed (mean $E_{FRET}$ = 0.83 ± 0.01 with Cdt1$^{N-649}$ compared to ~0.72 with unlabeled Cdt1, *Figure 4—figure supplement 1b*). Thus, if Cdt1 release is concomitant with OM interactions ending, the loss of half the acceptor fluorescence intensity due to Cdt1 release should occur simultaneously with the loss of FRET (*Figure 4b*, blue: $E_{FRET}$ panel). If Cdt1 release is disconnected from OM interactions ending, we would observe only a small reduction (from ~0.83 to ~0.72) in $E_{FRET}$ when Cdt1$^{N-649}$ is released.

We observed a clear correlation between the time that OM interactions end and the time of Cdt1 release on individual DNAs. We combined data from the first and second Mcm2-7 associations in 57 double-hexamer formation events, and observed 99 Cdt1 association and release events (*Figure 4c*, *Figure 4—figure supplement 3*). In the majority of these events (85/99), Cdt1 release is within experimental time resolution of the loss of OM-FRET (*Figure 4d*). In a smaller number of events (12/99), Cdt1 releases either immediately before or after OM-FRET ends (within ±7.2 s). We confirmed the loss of half the acceptor fluorescence cannot be consistently explained by photobleaching of the acceptor fluorophores on either Mcm2-7$^{2C-650}$ or Cdt1$^{N-649}$ (*Figure 4—figure supplement 2*). The distribution of intervals separating the end of the high $E_{FRET}$ state relative to the corresponding Cdt1 release is centered around 0 s (*Figure 4d*). This is true for both the first and second Cdt1 release events (*Figure 4—figure supplement 4*). Additionally, $E_{FRET}$ values are high in the 10-s interval before Cdt1 release, and low in the 10-s interval after Cdt1 release (*Figure 4—figure supplement 4*). Together, these data support a model in which Cdt1 release and the end of OM interaction are simultaneous or near-simultaneous, consistent with a connection between these events.

To test the hypothesis that Cdt1 release is coupled to the end of OM interaction, we monitored OM interaction in a mutant that is defective for Cdt1 release. We purified a version of the Mcm2-7$^{2C-649}$ protein with a mutation in the Mcm5-Mcm3 ATPase active site (Mcm5-R549A or 5RA). The Mcm2-7$^{5RA}$ mutant helicase is defective in Cdt1 release and helicase loading (*Coster et al., 2014*; *Kang et al., 2014*; *Ticau et al., 2017*). In reactions with ORC$^{5C-549}$ we observed associations of single Mcm2-7$^{2C-649,5RA}$ proteins (*Ticau et al., 2017*). Compared to wild-type Mcm2-7$^{2C-649}$ helicase, OM-FRET that began with Mcm2-7 arrival continued for much longer in the 5RA mutant helicase (average OM-FRET duration was 156 ± 14 s; *Figure 4e*). In all cases (128/128), we observed OM-FRET for the entire duration that both ORC$^{5C-549}$ (donor) and Mcm2-7$^{2C-649,5RA}$ (acceptor) were present, consistent with its duration being limited by photobleaching or loss of both proteins due to sliding off the DNA end. In contrast, in reactions with wild-type Mcm2-7$^{2C-649}$, the first OM interaction ends soon after the

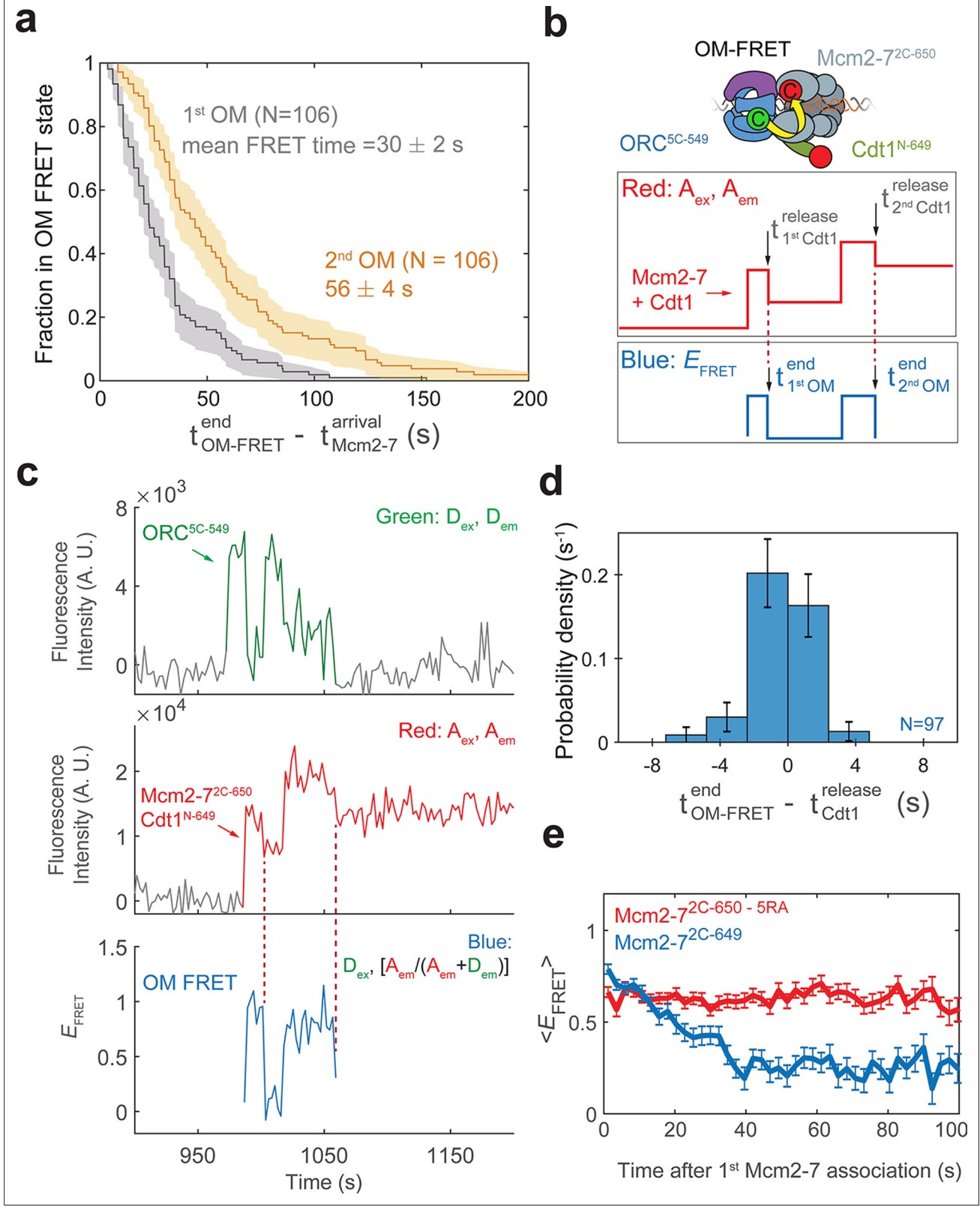

**Figure 4.** Cdt1 release is concomitant with the end of each OM interaction on individual DNAs. (**a**) Duration of the first (gray) and second (yellow) OM interactions after the association of the corresponding Mcm2-7 (from experiment in *Figure 1*) are plotted as survival functions. Shaded areas represent the 95% confidence intervals for each curve. (**b**) Experimental setup for observation of Cdt1 release and OM interactions in the same recording. Origin–recognition complex (ORC) and Mcm2-7 are labeled using the OM-FRET pair at $ORC^{5C-549}$ (donor) and $Mcm2-7^{2C-650}$ (acceptor). $Cdt1^{N-649}$ is also labeled

*Figure 4 continued on next page*

*Figure 4 continued*

with an acceptor fluorophore, but at a position that exhibits low $E_{FRET}$ with ORC$^{5C-549}$ (**Figure 4—figure supplement 1**). The example traces show the expected results if Cdt1 release is simultaneous with the end of the OM interaction. Each Mcm2-7$^{2C-650}$/Cdt1$^{N-649}$ association results in an increase in fluorescence in the acceptor-excited fluorescence record (red: $A_{ex}$, $A_{em}$ panel). Black arrows indicate the Cdt1$^{N-649}$ release, which results in loss of roughly half of the acceptor fluorescence. Each Mcm2-7$^{2C-650}$/Cdt1$^{N-649}$ arrival is accompanied by high $E_{FRET}$ due to OM interaction (blue: $E_{FRET}$ panel). The black arrows indicate the end of OM-FRET if Cdt1 release is concomitant with the end of OM interactions. (**c**) A representative trace showing the sequential association of two Mcm2-7$^{2C-650}$/Cdt1$^{N-649}$ molecules and OM-FRET. The red dashed lines indicate Cdt1-release times. Concentrations of labeled proteins in the reaction are 0.5 nM ORC$^{5C-549}$, 10 nM Mcm2-7$^{2C-649}$, and 10 nM Cdt1$^{N-649}$. *Figure 4—figure supplement 3* shows additional records. (**d**) Cdt1 release is correlated with the end of OM interactions. The duration of OM-FRET after Cdt1 release (± standard error [SE]) is plotted as a histogram. Data from the first and second Cdt1 releases are combined and plotted with a bin width of 2.4 s. We could observe Cdt1 association and release in 99/114 Mcm2-7 association events (Cdt1 was unlabeled in 15/114 events). Not shown are 2/99 events where OM interactions persisted much longer after the corresponding Cdt1 was released (56.8 and 179.8 s). (**e**) OM interaction times are extended when Cdt1 release is disrupted. Comparison of the population-average OM interaction $E_{FRET}$ values (± standard error of the mean [SEM]) at indicated times after the first Mcm2-7 association for Mcm2-7$^{2C-650,5RA}$ (which is defective in Cdt1 release; $N = 128$) and Mcm2-7$^{2C-649}$ ($N = 106$). In the experiment with Mcm2-7$^{2C-650,5RA}$, sample size was limited to the first half of the DNA locations (174/348), which had 128 Mcm2-7 association events with OM-FRET.

The online version of this article includes the following source data and figure supplement(s) for figure 4:

**Source data 1.** Primary data for histogram in *Figure 4d*.

**Figure supplement 1.** Cdt1$^{N-649}$ does not exhibit high $E_{FRET}$ with ORC$^{5C-549}$.

**Figure supplement 2.** Disappearance of Cdt1$^{N-649}$ fluorescence in the experiment in *Figure 4b, c* cannot be explained by photobleaching.

**Figure supplement 3.** Representative traces show Cdt1 release corresponds with the end of OM interactions on individual DNAs.

**Figure supplement 4.** Cdt1 release is concomitant with loss of OM $E_{FRET}$.

first Mcm2-7 arrival (mean 30 ± 2 s). Thus, the results from the Mcm2-7$^{5RA}$ mutant are consistent with the hypothesis that Cdt1 release is correlated with the end of OM interaction.

## Monitoring Mcm2-7–ORC interactions at a second interface

To investigate the protein movements that allow a single ORC to interact with both helicases, as predicted by the one-ORC (*Figure 3a*) and ORC-flip (*Figure 2a*) models, we monitored a second interface between Mcm2-7 and ORC during helicase loading. A recent structural study found evidence for the existence of a helicase-loading intermediate in which ORC interacts with the N-terminal face of Mcm2-7 (*Miller et al., 2019*). The authors referred to this structure as the 'MO complex' because of the opposite orientation of the ORC and Mcm2-7 proteins relative to their orientation in the O<u>CCM</u> structure that we have monitored up to this point (*Figure 5a*). To ask whether the MO interaction is formed during helicase loading, we modified ORC and Mcm2-7 at sites that are proximal (~40 Å apart) in the MO structure (*Figure 5b*) but are much further apart (~150 Å) in the OCCM (i.e., during the OM interaction studied in *Figures 1–4*). Specifically, ORC was labeled with a donor fluorophore at the Orc6 C-terminus (ORC$^{6C-549}$) and Mcm2-7 was labeled with an acceptor fluorophore at the Mcm3 N-terminus (Mcm2-7$^{3N-650}$). These proteins function equivalently to their unlabeled counterparts in ensemble helicase-loading experiments (*Figure 5—figure supplement 1*).

When ORC$^{6C-549}$ and Mcm2-7$^{3N-650}$ both colocalized with DNA, we observed periods of high $E_{FRET}$ (*Figure 5c*, *Figure 5—figure supplement 2*; blue: $D_{ex}$, [$A_{em}/(A_{em}+D_{em})$] panel). Because we labeled the MO interface, we will refer to periods of high $E_{FRET}$ arising from this set of probes as 'MO interactions'. In almost every observed double-hexamer formation event, we observed a period of MO interaction (94/95 events). These MO interactions occurred only once during helicase loading, and formation of the MO high $E_{FRET}$ state (MO-FRET) always anticipated the arrival of the second Mcm2-7 (94/94 events). Unlike the OM interactions we monitored previously, MO interactions do not begin simultaneously with the arrival of the first or second Mcm2-7. Instead, MO $E_{FRET}$ in the 10-s period immediately after first Mcm2-7 arrival was low, with an average value of 0.02 ± 0.02 (*Figure 5d*; top panel). In contrast, $E_{FRET}$ values in the 10-s interval just before second Mcm2-7 arrival were high with an average of 0.73 ± 0.01 (*Figure 5d*; middle panel). Interestingly, the MO interaction is lost rapidly after the second Mcm2-7 associates with DNA (average MO-FRET duration after second Mcm2-7 arrival was 9.7 ± 1.5 s). This short duration is reflected in the bimodal distribution of the $E_{FRET}$ values in the 10-s interval just after second Mcm2-7 arrival (*Figure 5d*; bottom panel).

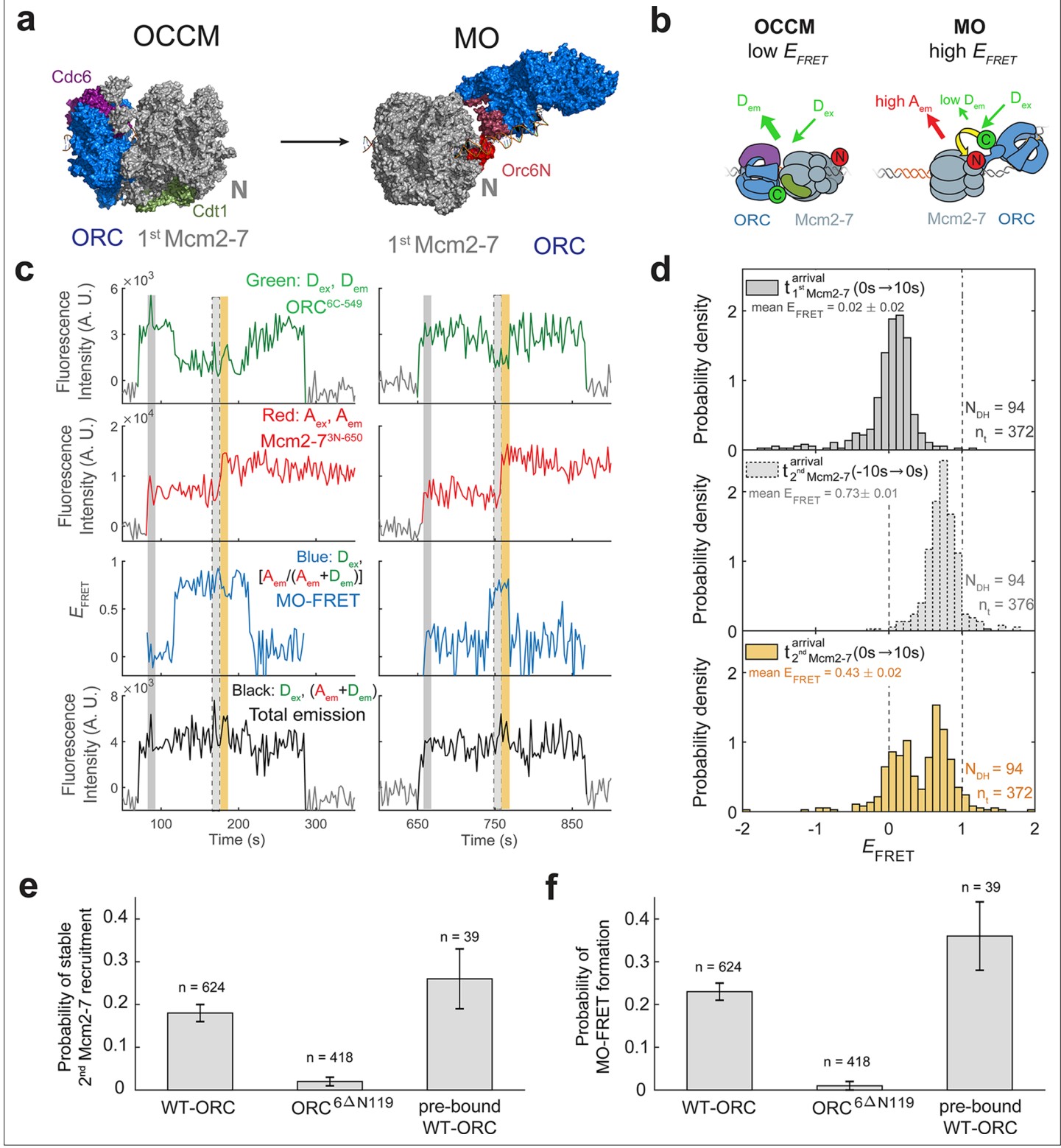

**Figure 5.** ORC forms a new 'MO' interaction interface with the first Mcm2-7 before recruiting the second Mcm2-7 helicase. (**a**) Cryo-EM structures of the OCCM (OM) and MO (PDB: 5V8F and 6RQC) show the inversion of ORC relative to the first Mcm2-7 helicase. (**b**) Schematic of ORC$^{6C-549}$ and Mcm2-7$^{3N-650}$ proteins in the context of the OCCM (left) and MO (right) structures. We predict that we would observe substantial FRET only in the MO but not the OCCM structure. (**c**) Representative traces showing ORC and Mcm2-7 associations with DNA and MO interaction $E_{FRET}$ during helicase loading. Donor-excited fluorescence record shows ORC$^{6C-549}$ association (green: $D_{ex}$, $D_{em}$ panel) and acceptor-excited fluorescence record shows Mcm2-7$^{3N-650}$ association

*Figure 5 continued on next page*

*Figure 5 continued*

(red: $A_{ex}$, $A_{em}$ panel). Formation of the MO interaction brings the donor and acceptor fluorophores in proximity resulting in increased FRET efficiency ($E_{FRET}$, blue: $D_{ex}$, [$A_{em}/ (A_{em} + D_{em})$] panel). The black: $D_{ex}$, ($A_{em} + D_{em}$) panel shows donor-excited total emission. Gray, gray-dash, and yellow highlight the three 10-s time intervals referenced in (**d**). Concentrations of proteins in the reaction are 0.5 nM ORC$^{6C-549}$, 15 nM Mcm2-7$^{3N-650}$, and 20 nM unlabeled Cdt1. *Figure 5—figure supplement 2* shows additional records. (**d**) Histograms of MO interaction $E_{FRET}$ values during three 10-s time intervals during helicase loading: 10 s after the first (top) and second (bottom) Mcm2-7 arrived and 10 s before the second Mcm2-7 arrived (middle). Examples of these intervals corresponding to the region highlighted in gray, yellow, and gray-dash in (**c**). The two dashed lines indicate $E_{FRET}$ values of 0 and 1. $N_{DH}$, number of double-hexamer formation events and $n_t$, number of signal points. $E_{FRET}$ values below −2 and above +2 were excluded (4/376, 0/376, and 4/376 signal points from the top, middle, and bottom histograms, respectively). (**e**) The N-terminal domain of ORC6 is required for second Mcm2-7 recruitment. The fraction (± standard error [SE]) of *n* first Mcm2-7$^{3N-650}$-binding events that resulted in stable recruitment of a second Mcm2-7$^{3N-650}$ is plotted for WT-ORC$^{6C-549}$, ORC$^{6C-549,6\Delta N119}$, and an experiment in which WT-ORC$^{6C-549}$/Cdc6 proteins are prebound to origin DNA. (**f**) The N-terminal domain of ORC6 is required for MO interaction. The fraction (± standard error [SE]) of *n* first Mcm2-7$^{3N-650}$-binding events that exhibited MO-FRET is plotted for WT-ORC$^{6C-549}$, ORC$^{6C-549,6\Delta N119}$, and prebound WT-ORC$^{6C-549}$/Cdc6 proteins.

The online version of this article includes the following source data and figure supplement(s) for figure 5:

**Source data 1.** Primary data for histogram in *Figure 5d* and graphs in *Figure 5e, f*.

**Figure supplement 1.** Fluorescent proteins used for detecting the MO interaction function in ensemble helicase-loading assays.

**Figure supplement 1—source data 1.** Primary data and quantification for graph in *Figure 5—figure supplement 1*.

**Figure supplement 2.** Representative traces showing MO interactions during helicase loading.

**Figure supplement 3.** First Mcm2-7 helicases that do not form MO complexes are rapidly released.

## Establishment of MO interactions stabilizes the first Mcm2-7 and is required for double-hexamer formation

To verify that the increases in $E_{FRET}$ that we refer to as MO-FRET represent formation of the previously described 'MO complex' (*Figure 5a*), we performed experiments with a mutant in Orc6 that disrupts MO formation (*Miller et al., 2019*). We purified a version of the ORC$^{6C-549}$ protein in which the N-terminal domain of Orc6 was deleted (ORC$^{6C-549,6\Delta N119}$). Ensemble helicase-loading assays using this protein show a strong reduction in successful helicase loading (*Miller et al., 2019*). In single-molecule experiments with ORC$^{6C-549,6\Delta N119}$ and Mcm2-7$^{3N-650}$, we nearly always observed association of a single Mcm2-7 helicase, but only 8/418 first Mcm2-7 association events went on to recruit a stable second Mcm2-7 (*Figure 5e*). Furthermore, ORC$^{6C-549,6\Delta N119}$ showed very few instances with MO-FRET (5/418 of first Mcm2-7 association events; *Figure 5f*). Both of these frequencies are much lower than what is observed with wild-type ORC$^{6C-549}$ (*Figure 5e, f*). These results are consistent with the MO-FRET signal being caused by formation of the MO complex and with formation of the MO interaction being critical for second Mcm2-7 recruitment.

Establishment of the MO interaction is likely to be a limiting step in helicase loading. Only a fraction (0.18 ± 0.02) of first Mcm2-7-binding events detected with wild-type ORC convert to stable second Mcm2-7 recruitment (*Figure 5e*, ORC$^{6C-549}$). Strikingly, this frequency is similar to the fraction of first Mcm2-7-binding events that have MO-FRET (0.23 ± 0.02; *Figure 5f*, ORC$^{6C-549}$). Consistent with the hypothesis that formation of the MO interaction is the step at which most unsuccessful helicase-loading attempts fail, formation of MO-FRET is a strong predictor of stable second Mcm2-7 association; 79 ± 3% (i.e., 0.18/0.23) of first Mcm2-7 association events with MO-FRET go on to recruit a stable second Mcm2-7.

A likely reason that MO formation is important for successful helicase loading is that the MO interaction stabilizes the first Mcm2-7 on the DNA. Consistent with this hypothesis, we find that when MO formation is inhibited in the context of the ORC$^{6C-549,6\Delta N119}$ mutant, the first recruited Mcm2-7 dissociates rapidly after ORC is released (*Figure 5—figure supplement 3a*). Under this condition, the lifetime of Mcm2-7 complexes after ORC dissociation are short, with a median lifetime of 11 ± 4 s (*Figure 5—figure supplement 3b*). This is not simply a consequence of the mutant ORC. Mcm2-7 complexes that are recruited by wild-type ORC but do not form an MO interaction show the same rapid release from DNA after ORC dissociation (median lifetime is 11 ± 2 s; *Figure 5—figure supplement 3c, d*). In comparison, Mcm2-7 complexes that form the MO interaction have longer lifetimes. A small fraction (0.21 ± 0.03) of the Mcm2-7 complexes that form MO interactions are released from

the DNA in contrast to the majority (0.79 ± 0.03) that go on to recruit a second Mcm2-7. For all the Mcm2-7 complexes that showed MO interactions we measured either (1) the time between the onset of the MO interaction and the Mcm2-7 departure from the DNA, or (2) the time between the onset of the MO interaction and the arrival of the second Mcm2-7. We note that because the latter measurement ends not with Mcm2-7 release but second Mcm2-7 arrival the resulting measured Mcm2-7 lifetimes are an underestimate. Nevertheless, the median lifetime for all first Mcm2-7 complexes that formed MO interactions was 35 ± 7 s, compared to the median lifetime of 11 ± 2 s following ORC departure for Mcm2-7 complexes that did not form the MO interaction (*Figure 5—figure supplement 3d*). Finally, we found that establishing MO interactions extends ORC dwell times on DNA (*Figure 5—figure supplement 3e-f*). Together, these findings are consistent with MO complex formation stabilizing both ORC and the first Mcm2-7 helicase on the DNA.

## Prebound ORC–Cdc6 can form double hexamers without additional ORC in solution

To further confirm that one ORC can mediate MO formation and recruit both Mcm2-7 helicases, we performed a staged helicase-loading reaction. We incubated slide-attached origin DNA with ORC$^{6C-549}$/Cdc6, washed away unbound proteins and then performed the single-molecule helicase-loading reaction by adding all the helicase-loading proteins except ORC (see Materials and methods). Although the reaction did not contain any additional ORC, we observed second Mcm2-7 recruitment and MO-FRET in a fraction of first Mcm2-7 association events (0.26 ± 0.07 and 0.36 ± 0.08, respectively; *Figure 5e, f*, prebound ORC). Importantly, these ratios are similar to those seen for the unstaged reaction (*Figure 5e, f*, WT-ORC). Consistent with our observations with the unstaged reaction, MO-FRET anticipated the recruitment of the second Mcm2-7 in all double-hexamer formation events (10/10 events) with prebound ORC$^{6C-549}$/Cdc6. Thus, helicase loading proceeds similarly even when free ORC is removed prior to second Mcm2-7 recruitment.

## Cdt1 release coordinates the ORC flip on DNA

To examine more directly the order of events surrounding MO formation, we monitored Cdt1 release and MO-FRET in a single experiment. The SM helicase-loading experiment was performed with the MO-FRET pair ORC$^{6C-549}$ and Mcm2-7$^{3N-650}$. Additionally, Cdt1 was labeled at its C-terminus with an acceptor fluorophore (Cdt1$^{C-650}$). Similar to the approach in *Figure 4b*, association of an Mcm2-7$^{3N-650}$/Cdt1$^{C-650}$ results in the association of two red fluorophores (*Figure 6a*, red: $A_{ex}$, $A_{em}$ panel). The subsequent loss of half the acceptor fluorescence intensity soon after Mcm2-7/Cdt1 arrival is indicative of Cdt1 release. Proximity between donor-labeled ORC$^{6C-549}$ and acceptor-labeled Mcm2-7$^{3N-650}$ results in one period of increased $E_{FRET}$ (MO interaction) that starts before the recruitment of the second Mcm2-7. Although Cdt1$^{C-650}$ is also labeled with an acceptor fluorophore, ORC$^{6C-549}$ exhibits very low FRET with Cdt1$^{C-650}$ (average $E_{FRET}$ is 0.10 ± 0.01; *Figure 6—figure supplement 1*). Thus, we can monitor both MO interactions and Cdt1 arrival and departure in these experiments.

Kinetic analysis of MO-FRET onset indicates that the MO intermediate is formed shortly after the first Cdt1 is released and the OM interaction is lost. We analyzed 78 events with stable second Mcm2-7 recruitment and detectable first Cdt1 release (e.g., *Figure 6b*). We confirmed that the loss of half the acceptor-excited acceptor fluorescence, which we infer as Cdt1 release, is unlikely to be due to photobleaching of the acceptor fluorophore on Cdt1$^{C-650}$ (*Figure 6—figure supplement 2*). In 76/78 events, MO-FRET is either established simultaneously with release of the first Cdt1 (within the time resolution of the assay, i.e., ±2.4 s; see *Figure 1—figure supplement 2b*) or occurs after the first Cdt1 is released. Strikingly, MO interactions began unambiguously after Cdt1 release (i.e., delayed by >2.4 s) in a majority of these events (59/78 events; *Figure 6c*). The average time to MO-FRET after first Cdt1 release is 8.3 ± 2.0 s indicating that establishment of MO-FRET is a separate, subsequent event from the end of the first OM-FRET, which is simultaneous with the first Cdt1 release (*Figure 4d*). Consistent with the first Cdt1 release occurring prior to formation of the MO interaction, MO-$E_{FRET}$ values are mostly low in periods when the first Cdt1 is associated and high after the first Cdt1 is released (*Figure 6—figure supplement 3*).

To further establish the order of first Cdt1 release and formation of the MO interaction, we monitored the formation of MO interactions in a mutant that is defective for Cdt1 release. To this end we used the same Mcm2-7$^{5RA}$ mutant that is defective for Cdt1 release and second Mcm2-7 recruitment

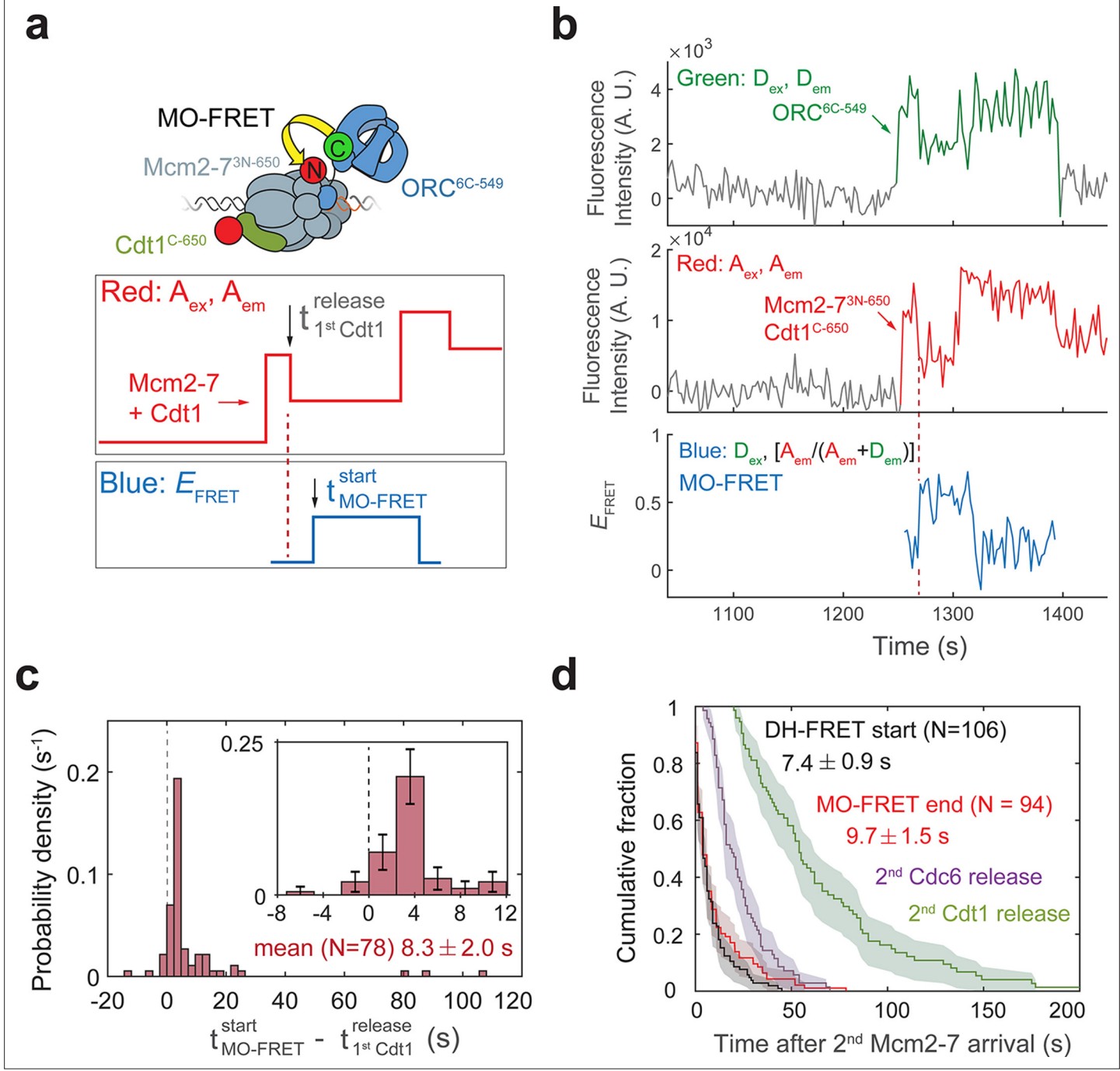

**Figure 6.** Cdt1 release occurs before MO formation. (**a**) Experimental setup for observation of Cdt1 release and MO interactions on the same DNA molecule. Origin–recognition complex (ORC) and Mcm2-7 are labeled using the MO-FRET pair ORC$^{6C-549}$ (donor) and Mcm2-7$^{3N-650}$ (acceptor), respectively. Cdt1$^{C-650}$ is also labeled with an acceptor fluorophore, but at a position that exhibits low $E_{FRET}$ with ORC$^{6C-549}$ (*Figure 6—figure supplement 1*). The panels below show the predicted traces if Cdt1 release occurs before MO formation during a helicase-loading event. Each Mcm2-7$^{3N-650}$/Cdt1$^{C-650}$ association results in an increase in fluorescence in the acceptor-excited fluorescence record (red: $A_{ex}$, $A_{em}$ panel). The black arrow indicates the release of Cdt1$^{C-650}$ that results in a decrease in acceptor-excited fluorescence roughly half the size of the prior increase. (**b**) A representative trace showing the sequential association of two Mcm2-7$^{3N-650}$/Cdt1$^{C-650}$ molecules and MO-FRET. The red dashed line indicates the first Cdt1-release time. *Figure 6—figure supplement 3a* shows additional records. Concentrations of proteins in the reaction are 0.5 nM ORC$^{6C-549}$, 10 nM Mcm2-7$^{3N-650}$, and 20 nM Cdt1$^{C-650}$. (**c**) Release of the first Cdt1 anticipates the onset of MO interactions. The start time of MO-FRET after the first Cdt1 release is plotted as a histogram with bin size of 2.4 s. Dashed line indicates a time difference of zero between first Cdt1 release and MO-FRET start time. Inset: magnified view (± standard error [SE]). (**d**) MO interactions end rapidly after arrival of the second Mcm2-7 with a timing that matches that of double-hexamer formation.

*Figure 6 continued on next page*

*Figure 6 continued*

Cumulative distributions of four events after second Mcm2-7 arrival are plotted: the time to end of MO interaction (red), start of double-hexamer interactions detected via FRET (black) (*Ticau et al., 2015*), and release of the second sets of Cdc6 and Cdt1 molecules (*Ticau et al., 2015*). Shaded areas represent the 95% confidence intervals for each curve. Mean ± SE are given for the red and black datasets.

The online version of this article includes the following source data and figure supplement(s) for figure 6:

**Source data 1.** Primary data for histogram in *Figure 6c*.

**Figure supplement 1.** Cdt1$^{C-650}$ does not exhibit high $E_{FRET}$ with ORC$^{6C-549}$.

**Figure supplement 2.** Disappearance of Cdt1$^{C-650}$ fluorescence in the experiment in *Figure 6a, b* cannot be explained by photobleaching.

**Figure supplement 3.** The high MO $E_{FRET}$ state is established rapidly after the first Cdt1 is released.

**Figure supplement 4.** The first Cdt1 release anticipates formation of the MO interaction.

---

but now labeled to detect MO interactions. In reactions with ORC$^{6C-549}$ and Mcm2-7$^{3N-650,5RA}$ molecules, we observed only single Mcm2-7 association in 162/163 Mcm2-7-binding events (*Figure 6—figure supplement 4a*). Consistent with Cdt1 release being required for MO formation, only 3/163 events showed MO interactions (fraction of events is 0.02 ± 0.01). This is a much lower fraction than is observed for the wild-type protein (0.23 ± 0.02, *Figure 6—figure supplement 4b*). Together, these data support a model in which Cdt1 release anticipates formation of the MO interaction.

What drives the dissolution of the MO complex? There are three temporally separable events known to occur after arrival of the second Mcm2-7: (1) initiation of double-hexamer interactions between the two helicases (*Ticau et al., 2015*); (2) release of the second Cdc6 (*Ticau et al., 2015*); and (3) three typically simultaneous events of release of the second Cdt1, closing of the second Mcm2-7 ring, and release of ORC (*Ticau et al., 2017*). We compared the distribution of times of loss of MO interaction relative to the arrival of the second Mcm2-7 to the previously determined times of each of these events (*Figure 6d*). The distribution of second Cdc6 and second Cdt1-release times are clearly distinct from the times of MO dissolution. In contrast, we observed a similar distribution for the end of the MO interaction after arrival of the second Mcm2-7 (*Figure 6e*, 9.7 ± 1.5 s) relative to the time required for initial double-hexamer interactions after second Mcm2-7 arrival (*Ticau et al., 2015*). These data strongly suggest that formation of double-hexamer interactions between the first and second helicases accompany a structural reconfiguration that ends the MO interaction.

## Discussion

By monitoring interactions between ORC and Mcm2-7 through two separate interfaces, the single-molecule experiments described here provide unique insights into helicase loading. We demonstrate that similar, if not identical, interactions between DNA-bound ORC and the C-terminal regions of Mcm2-7 occur during recruitment of the two helicases during double-hexamer formation (OM interactions, *Figures 1 and 2*). Importantly, we show that one ORC is sufficient to efficiently recruit both helicases (*Figure 3*) but that in rare cases separate ORC molecules may recruit the two helicases. Between the two helicase recruitment events, ORC forms a distinct interface with the N-terminal region of the first recruited Mcm2-7 (MO interaction, *Figure 5*) that mediates retention of ORC on the DNA and is required for recruitment of the second Mcm2-7. Finally, our experiments define the order for Cdt1 release and establishment of double-hexamer interactions relative to OM and MO interactions (*Figures 4 and 6*). Overall our findings reveal a highly coordinated series of events ensuring the formation of the correct head-to-head conformation of the loaded helicases.

### A single-ORC model for helicase loading

Based on our findings, we propose a detailed model for helicase loading by one ORC (*Figure 7*). ORC-Cdc6 bound to origin DNA recruits the first Cdt1-bound-Mcm2-7 helicase via the first OM interaction (*Figure 7*; OC$^6$C$^1$M$_1$ state). This complex is maintained until the ATP-hydrolysis-dependent releases of Cdc6 and then Cdt1, forming the OM$_1$ state. Following release of the first Cdt1, ORC releases from its initial DNA- and Mcm2-7-binding sites to form a distinct interaction with the N-terminal region of the first Mcm2-7 (pre-M$_1$O state). This tethered conformation prevents ORC from releasing into solution as it exchanges DNA-binding sites. As a consequence of the ORC inversion, the two helicases

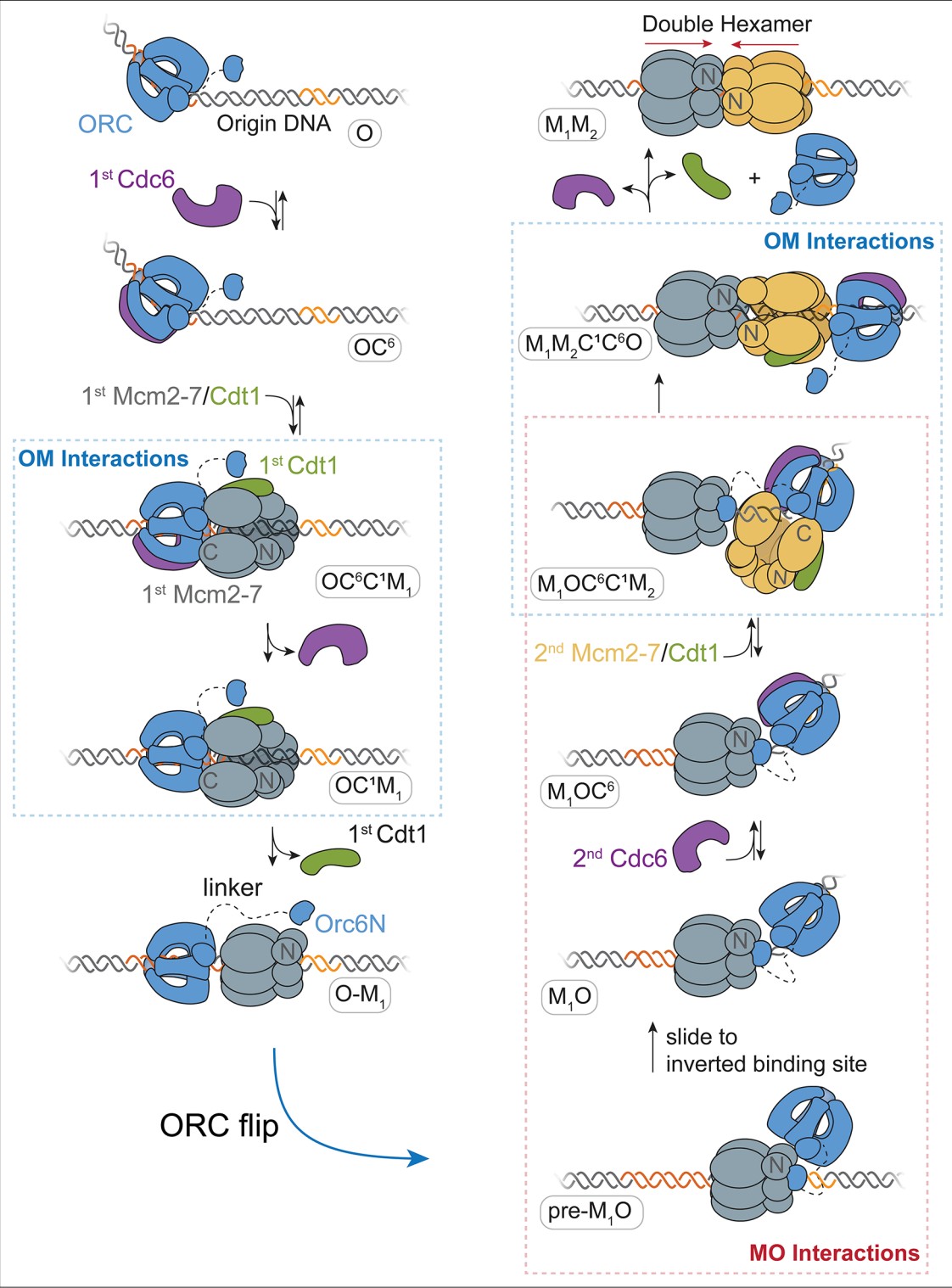

**Figure 7.** A model for helicase loading mediated by one ORC molecule. Designations in ovals are unique names for each intermediate using the abbreviations O, ORC; $C^6$, Cdc6; $C^1$, Cdt1; $M_1$, first Mcm2-7; and $M_2$, second Mcm2-7. Dashed curve represents the linker that connects the N- and C-terminal domains of Orc6. Orange and yellow DNA segments are sequence-specific ORC-binding sites, see text for details. Note that the ORC and Mcm2-7 proteins are not interacting in the O–$M_1$ complex.

The online version of this article includes the following figure supplement(s) for figure 7:

**Figure supplement 1.** A model for helicase loading mediated by two separate ORC molecules.

are recruited in opposite orientations ($M_1OC^6C^1M_2$ state) and rapidly interact to form a head-to-head Mcm2-7 double hexamer ($M_1M_2$).

The model we present reconciles previous observations that seemed contradictory. A single ORC making similar OM interactions during recruitment of both Mcm2-7 complexes is consistent with previous observations that Mcm2-7 mutations in this interface interfere with recruitment of both the first and second helicases (*Coster and Diffley, 2017*; *Frigola et al., 2013*). Further, the proposal that ORC recruits both helicases by rebinding DNA in an inverted orientation corresponds with a requirement for two ORC-binding sites at origins (*Coster and Diffley, 2017*). We show that the MO interaction is consistently present during recruitment of the second Mcm2-7. This finding is consistent with structural studies suggesting that an ORC molecule engaged with the N-terminal region of the first Mcm2-7 is responsible for recruiting the second Mcm2-7 (*Miller et al., 2019*). Because the MO complex holds ORC in close proximity to the first Mcm2-7 as the second Mcm2-7 is recruited, this finding explains the rapid formation of double-hexamer interactions after second Mcm2-7 arrival (*Ticau et al., 2015*). Finally, our observation that a single ORC molecule can guide helicase loading is consistent with previous single-molecule studies and time-resolved cryo-EM studies, both of which observed predominantly one ORC engaged with DNA during helicase loading (*Miller et al., 2019*; *Ticau et al., 2015*).

Our experiments reveal that the events in helicase loading guided by a single ORC are highly coordinated. The first OM interaction begins simultaneously with arrival of the first Mcm2-7/Cdt1 and ends when the first Cdt1 is released. The MO interaction is formed shortly after the first Cdt1 is released and the OM interaction ends (mean 8.7 ± 2.0 s after Cdt1 release), and anticipates recruitment of the second Mcm2-7. Consistent with this ordering, preventing Cdt1 release with a Mcm2-7 ATPase mutant extends the duration of OM interactions (average OM duration after first Mcm2-7 arrival is 30 ± 2 s for wild type and 156 ± 14 s for the Mcm2-7$^{5RA}$ mutant) and prevents formation of MO interactions. The second OM interaction begins simultaneously with the second Mcm2-7 arrival ($M_1OC^6C^1M_2$ state). We infer that establishment of the MO interaction is required to recruit the second Mcm2-7 helicase based on two observations. First, we consistently observe that the onset of the MO interaction anticipates recruitment of the second Mcm2-7. Second, disrupting the MO interaction using a mutant version of ORC prevents second Mcm2-7 recruitment (*Figure 5*). Finally, the temporal connection between the formation of double-hexamer interactions and the end of the MO interactions (*Figure 6d*), strongly suggests that formation of double-hexamer interactions breaks the MO interaction. This supports the existence of two complexes with the same set of proteins but distinct ORC–Mcm2-7 and Mcm2-7–Mcm2-7 interactions (*Figure 7*, $M_1OC^6C^1M_2$ and $M_1M_2C^1C^6O$ states). The requirement to form the MO complex to recruit the second Mcm2-7 combined with the requirement to disrupt the MO complex to establish double-hexamer interactions together provide an ordered selection to ensure that successfully recruited second Mcm2-7 helicases go on to form double hexamers.

We propose that a single ORC molecule uses a tether to the first helicase to remain DNA associated while ORC locates and binds a second inverted DNA site. This mechanism is supported by the observation that ORC is not bound to the primary ORC-binding site on DNA in a structure of the MO complex (*Miller et al., 2019*). Although protein flipping has not been reported in other steps of DNA replication, multiple DNA-binding proteins have been shown to switch orientations on DNA (*Ganji et al., 2016*). In these previously reported systems, protein inversion is proposed to occur via dissociation into solution, followed by rapid reorientation and rebinding resulting in the flipped protein binding the same (quasi-)palindromic DNA site (*Marklund et al., 2020*; *Sasnauskas et al., 2011*), or a closely spaced inverted binding site (*Ryu et al., 2014*). The ORC gymnastics observed here however are distinguished from other protein flipping events by the use of a protein tether between ORC and the first Mcm2-7 helicase.

The full role of ATP binding and hydrolysis during helicase loading remains to be determined. We have shown that a mutation in Mcm5 predicted to eliminate ATP hydrolysis at one of the six Mcm2-7 ATPases prevents Cdt1 release (*Ticau et al., 2017*) and OM complex dissolution (*Figure 4e*). Whether ATP hydrolysis by other Mcm2-7 ATPases catalyze the same or other events remains to be determined. ORC and/or Cdc6 ATP hydrolysis could also be involved in the release of ORC from the DNA. ORC binding to origin DNA is stimulated by ATP but not ADP (*Bell and Stillman, 1992*), suggesting ORC ATP hydrolysis could facilitate release of ORC from its initial binding site. Cdc6 completes the ORC–Cdc6 protein ring around the DNA (*Yuan et al., 2017*) and Cdc6 ATP hydrolysis is required to

release Cdc6 (*Coster et al., 2014*; *Kang et al., 2014*), suggesting that this hydrolysis event also is required for ORC DNA release.

## Origin structure and helicase loading

A model in which the MO complex is required to recruit the second Mcm2-7 is consistent with origin architecture in budding yeast, wherein two ORC-binding sites with different affinities are required for helicase-loading in vivo. Previous studies have shown that in addition to the primary ORC-binding site (ARS consensus sequence or ACS), natural origins consistently have at least one (and sometimes multiple) additional weak, inverted ORC-binding sites referred to as B2 elements (*Bell and Stillman, 1992*; *Marahrens and Stillman, 1992*; *Palzkill and Newlon, 1988*; *Rao et al., 1994*; *Wilmes and Bell, 2002*). Interestingly, even though two oppositely oriented strong ORC-binding sites support origin function both in vitro and in vivo (*Coster and Diffley, 2017*), natural origins have selected against this architecture (*Chang et al., 2011*; *Marahrens and Stillman, 1992*; *Palzkill and Newlon, 1988*; *Rao et al., 1994*). This selection argues against a model in which ORC is prebound at both sites. Indeed, when B2 elements have been mapped they are frequently located at positions in which the presence of the OCCM (OC$^6$C$^1$M$_1$ in *Figure 7*) would clash with ORC binding at B2 (*Chang et al., 2011*; *Marahrens and Stillman, 1992*; *Rao et al., 1994*). Along these lines, in the small number of two ORC events in our studies, we observe second ORC associations occur after the first, but before the second, Mcm2-7 arrives. Thus, we propose that, whether one or two ORCs mediate loading, the ORC mediating second Mcm2-7 recruitment would only stably associate with B2 after the first Mcm2-7 is recruited. Such a mechanism would ensure that these sites are only bound once the first Mcm2-7 is recruited, thus driving the reaction towards double-hexamer formation and preventing two ORC proteins independently recruiting 'first' Mcm2-7 complexes.

The relative placement of the two ORC-binding sites might influence the coordinated series of events we observe in helicase loading. Although the two oppositely oriented ORC-binding sites are very close in the *ARS1* origin used in our experiments, the distance between these sites is variable across natural origins (*Chang et al., 2011*; *Palzkill and Newlon, 1988*). We speculate that the pre-M$_1$O slides in search of a second, inverted ORC-binding site (*Figure 7*) and recruits the second Mcm2-7 only once an inverted ORC-binding site is found. As described above, for a number of origins for which B2 elements have been identified (including *ARS1*) this sliding would be required to expose the B2 element (in *Figure 7*, sliding to the left). A number of helicase-loading intermediates slide in vitro (*Sánchez et al., 2021*; *Scherr et al., 2021*), and experiments placing barriers to sliding at *ARS1* support a role for sliding to identify second inverted ORC-binding sites (*Coster and Diffley, 2017*; *Warner et al., 2017*). Consistent with the argument that second Mcm2-7 recruitment only occurs when the sliding pre-M$_1$O finds a second inverted ORC-binding site, helicase loading does not occur at artificial origins with a single strong ORC-binding site (*Coster and Diffley, 2017*).

## Comparison of one-ORC and twoORC mechanisms for helicase loading

Our data also demonstrate that two distinct ORC molecules are present in a small subset of helicase-loading events and provide information about the mechanism of these events. One insight comes from the analysis of first Mcm2-7 DNA retention in the absence of MO formation. Analyses of helicase loading with the ORC$^{6\Delta N119}$ mutant and first Mcm2-7 helicases that fail to form the MO interaction show that in both cases the Mcm2-7 molecules are rapidly released from the DNA after ORC release (*Figure 5—figure supplement 3*). This finding indicates that the first Mcm2-7 is unstable if the MO complex is not formed and that there would be only a short-time window for a second ORC molecule to form an MO complex before the first Mcm2-7 is released. It is also interesting to consider how the MO complex is formed by a second ORC. There are two possible ways for a second ORC to interact with the first Mcm2-7 (*Figure 7—figure supplement 1*). It is possible that the second ORC is recruited out of solution by MO-like interactions with the first Mcm2-7 and then identifies an appropriate B2 DNA-binding site. Alternatively, ORC could bind at the low-affinity B2 site first and then form the MO interaction with the first Mcm2-7. In both cases, there would be a short-time window for the second ORC to form the MO before the first Mcm2-7 is released. Analysis of the two-ORC events in our assay that detects the MO complex suggests both mechanisms are used.

It is likely that both one- and two-ORC mechanisms are used in vivo and it will be difficult to determine which mechanism is used more frequently. The low frequency of two-ORC events we observe

(see *Figure 3*) suggests that the first ORC has a higher probability of forming the MO complex through a helicase-tethered flip than a second ORC, consistent with the first ORC already being associated with the first Mcm2-7. On the other hand, our single-molecule experiments are necessarily performed at low ORC concentrations. Given the short-time window for a second ORC to act, higher ORC concentrations might increase the frequency of two-ORC events. Finally, we note that it is difficult to determine the free ORC concentration in vivo, particularly given the ability of ORC to interact nonspecifically with AT-rich DNA. Interestingly, metazoan ORC can sequester into a separated liquid phase (*Hossain et al., 2021*; *Parker et al., 2019*). Because ORC phase separation requires DNA and may involve other proteins in vivo, it is not clear whether or not the free ORC concentration would be elevated in the predicted condensates. Nevertheless, consistent with ORC being limiting in cells, in vivo footprinting studies show that only a subset of the origin-associated stronger ACS-binding sites are occupied by ORC in cells (*Belsky et al., 2015*). This finding suggests that the much weaker B2-binding sites are unlikely to stably bind ORC without additional interactions. Thus, we hypothesize that if a second ORC is involved in vivo it would only be recruited after the first Mcm2-7 is origin associated.

## Cdt1 release coordinates the transition between OM and MO complexes

Our data show that release of the first Cdt1 is a critical regulatory step in helicase loading. There are two potential explanations for the simultaneous end of OM interactions and Cdt1 release observed here. Cdt1 binding could be required for OM interactions such that Cdt1 release reduces or eliminates these interactions. Consistent with this hypothesis, Cdt1 binding has been proposed to alleviate an autoinhibitory function of Mcm2-7, thereby allowing interaction with ORC-CdcDNA (*Fernández-Cid et al., 2013*). A second possibility is that closure of the Mcm2-5 gate, which is concomitant with Cdt1 release (*Ticau et al., 2017*), induces a conformational change in the Mcm2-7 complex that disrupts the OM interaction. It is intriguing to note that ORC is frequently released from the DNA simultaneously with the second Cdt1 release (*Ticau et al., 2015*), suggesting that in addition to breaking the OM interaction, release of the first Cdt1 may stimulate ORC release from DNA.

Release of the first Cdt1 and the concomitant closure of the Mcm2-5 gate around DNA likely create a new interaction site on Mcm2-7 for Orc6 (*Miller et al., 2019*; *Ticau et al., 2017*). The N- and C-terminal domains of Orc6 (Orc6N and Orc6C) interact with Cdt1 in vitro (*Chen and Bell, 2011*). We propose that once the first Cdt1 is released, Orc6N is able to reach over the first Mcm2-7 and establish a tether with the Mcm2 N-terminal region (*Miller et al., 2019*). In this model, the extensive linker (~160 amino acids) between the Orc6N and Orc6C serves as flexible leash that facilitates a search for this new Orc6N-binding site (*Chen and Bell, 2011*). Since ORC is not Cdc6 bound at this stage (OM$_1$ state), ORC forms an open ring around DNA and can release from its primary DNA-binding site. Additional events that could contribute to ORC release from its initial high-affinity-binding site, include the conformational changes driving dissolution of the OM complex and changes in ORC conformation upon binding to the N-terminal region of Mcm2-7. The closed Mcm2-5 gate in the M$_1$O facilitates contacts between Orc6C and the Mcm5 N-terminal region such that the two domains of Orc6 form a latch across the Mcm2-5 gate (*Miller et al., 2019*), a likely mechanism for the MO complex stabilizing DNA association of the first Mcm2-7 after loss of the OM interaction (*Figure 5—figure supplement 3*). We show that Orc6N is required to form the MO interaction, and that release of the first Cdt1 and formation of the MO interaction occur in close succession. However, we could not determine the timing of establishment of the Orc6N tether (as in the pre-M$_1$O intermediate) relative to the end of the first OM interaction (or the first Cdt1 release), as fluorophore modification of Orc6 N-terminal domain interfered with helicase loading.

## Implications for origin licensing at non-sequence defined origins

A single-ORC mechanism for helicase loading has potential implications for DNA replication initiation in higher eukaryotes. Although the helicase-loading proteins from budding yeast are highly conserved, ORC does not exhibit sequence-specific binding in higher eukaryotes where origins are not defined by specific DNA sequences (*Bleichert et al., 2017*; *Prioleau and MacAlpine, 2016*; *Vashee et al., 2003*). A mechanism in which two separate ORC molecules guide loading of the two helicases would require two independent nonspecific ORC-binding events. Each single Mcm2-7 helicase would then

have to find an oppositely oriented helicase to complete helicase loading. In contrast, helicase loading guided by a single ORC that performs an inversion places fewer constraints on where helicases can be loaded and could still be supported with low ORC concentrations. Additionally, a full round of helicase loading completed by a single ORC could prevent non-productive loading of single helicases that cannot support bidirectional replication (*Champasa et al., 2019*; *Miller et al., 2019*).

## Materials and methods
### Protein purification and labeling

Wild-type ORC and Cdc6 were purified as described previously (*Frigola et al., 2013*). We used a N- and C-terminal protein modifications to fluorescently label ORC (-LPETGG at the C-terminus of Orc5 and Orc6, Ubiquitin-GGG-Flag at the N-terminus of Orc1), Mcm2-7 (LPETGG at the C-terminus of Mcm2, Ubiquitin-GGG-Flag at the N-terminus of Mcm4, 3xFlag-TEV[ENLYFQ/G]-GG at the N-terminus of Mcm3), and Cdt1 (Ubiquitin-GGG tag at the N-terminus, LPETGG tag at the C-terminus). The ubiquitin (in vivo) and 3xFlag-TEV (using TEV protease, NEB) fusions were removed to reveal three N-terminal glycines required for N-terminal sortase-mediated labeling. The peptides $NH_2$-CHHHHHHHHHHLPETGG-COOH and $NH_2$-GGGHHHHHHHHHHC-COOH were used for N- and C-terminal labeling, respectively, and will be referred to as the N-peptide and C-peptide. The N- and C-peptides were labeled with maleimide-derivatized DY549P1 (Dyomics), DY649P1 (Dyomics), Dylight550 (Thermo-scientific), Dylight650 (Thermo-scientific), or Cy3B (Cytiva) via cystine–maleimide conjugation as described previously (*Ticau et al., 2015*). Sortase was used to couple the fluorescently labeled peptide to the N- or C-terminus of the helicase-loading proteins as described below. The peptide-coupled proteins were separated from uncoupled proteins using Ni-NTA Agarose (Qiagen).

### Note on fluorophore-modified protein nomenclature

We use a shorthand for the site of fluorescent modification and the fluorophore to indicate various labeled proteins (e.g. $ORC^{5C-549}$). The numerical superscripts 549 and 649 indicate proteins coupled to the fluorophores DY549P1 and DY649P1, respectively. Similarly, the superscripts 550 and 650 indicate the fluorophores Dylight550 and Dylight650. DY549P1 and Dylight550 have almost identical spectral properties, and the same is true for DY649P1 and Dylight650. The numerical subscripts are preceded by the site of fluorescent labeling on the protein. Thus, $ORC^{5C-549}$ indicates ORC labeled at the C-terminus of Orc5 with the DY549P1 dye.

### Preparation of labeled $ORC^{5C-549}$, $ORC^{5C-Cy3B}$, and $ORC^{6C-549}$

*S. cerevisiae* (W303 background) strains ySG10 (CBP-Orc1, Orc2-4, $Orc^{5C-LPETGG}$, Orc6) and ySG40 (CBP-Orc1, Orc2-5, $Orc^{6C-LPETGG}$) were grown to $OD_{600}$ = 1.2 in 8 l of YEP supplemented with 2% glycerol (wt/vol) at 30°C. Cells were arrested in G1 using α-factor (100 ng/ml) and ORC expression was induced using 2% galactose (wt/vol). After 6 hr, cells were harvested and sequentially washed with 150 ml ORC wash buffer (25 mM HEPES–KOH pH 7.6, 1 M sorbitol) and ORC lysis buffer (25 mM HEPES–KOH pH 7.6, 0.05% NP-40, 10% glycerol + 0.5 M KCl). The washed pellet was resuspended in approximately 1/3 of packed cell volume of ORC lysis buffer containing cOmplete Protease Inhibitor Cocktail Tablet (1 tablet per 25 ml total volume; Roche) and frozen dropwise in liquid nitrogen.

Frozen cells were lysed in a SamplePrep freezermill (SPEX) and the lysate was clarified by ultracentrifugation in a Type 70 Ti rotor at 45 krpm for 1 hr at 4°C. The supernatant was supplemented with 2 mM $CaCl_2$ and applied to 2.4 ml bed volume (BV) calmodulin-affinity resin (Agilent) pre-equilibrated in ORC lysis buffer containing 2 mM $CaCl_2$. The supernatant and resin were incubated with rotation for 1.5 hr at 4°C. The resin was collected on a column and the flow-through was discarded. The resin was then washed with 10 BV of buffer A (50 mM HEPES–KOH pH 7.6, 5 mM $Mg(OAc)_2$, 10% glycerol) supplemented with 0.2 M KCl, 0.02% NP-40, and 2 mM $CaCl_2$. ORC was eluted with 5 BV buffer A containing 0.2 M KCl, 0.02% NP-40, 1 mM EDTA, and 2 mM egtazic acid(EGTA), with 15 min incubations before each elution. Elutions containing ORC (typically fractions 2–4) were pooled and applied to 1 ml BV SP Sepharose Fast Flow (Cytiva) pre-equilibrated with buffer A containing 0.2 M KCl and 0.02% NP-40. The resin was washed with 10 BV of buffer A containing 0.2 M KCl and 0.02% NP-40. ORC was eluted with 5 BV of Buffer A containing 500 mM KCl, 0.02% NP-40. Note that for the ORC protein with the Ubiquitin-GGG tag (the N-terminal

methionine is replaced with the ubiquitin followed by three glycines), the N-terminal ubiquitin is cleaved off in vivo resulting in three glycines at the N terminus of Orc1. Starting with 8 l of cells, the yield was typically 3 mg of ORC.

About 3 nmol of purified ORC (1.5 mg, <0.9 ml) was incubated with equimolar amount of Srt5° evolved sortase (*Chen et al., 2011*; *Ticau et al., 2015*) and $CaCl_2$ was added to a final concentration of 5 mM. This was mixed with 100 nmol of either C-peptide labeled with DY549-P1 (for $ORC^{5C-549}$ and $ORC^{6C-549}$) or C-peptide labeled with Cy3B (for $ORC^{5C-Cy3B}$). The reaction was incubated at room temperature for 15 min, and then quenched with 20 mM EDTA. After dye coupling, the reaction was applied to a Superdex 200 10/300 gel filtration column equilibrated in buffer B (50 mM HEPES–KOH pH 7.6, 5 mM $Mg(OAc)_2$, 10% glycerol, 0.3 M potassium glutamate [KGlu], 0.02% NP-40) containing 10 mM imidazole. Peak fractions containing peptide-coupled ORC were pooled and incubated with 0.3 ml of Ni-NTA Agarose Resin (Qiagen) pre-equilibrated in buffer B with 10 mM imidazole, for 1.5 hr with rotation at 4°C. The flow-through was discarded and the resin was washed sequentially with 3 ml buffer B with 15 mM imidazole and buffer B with 25 mM imidazole. Peptide-coupled ORC was eluted using buffer B with 0.3 M imidazole. Peak fractions were pooled, aliquoted, and stored at −80°C.

## Preparation of labeled ORC$^{6C-549,6\Delta N119}$

The *S. cerevisiae* strain ySG41 (CBP-Orc1, Orc2-5, 3x-Flag-ΔN119-Orc6$^{C-LPETGG}$) was grown to $OD_{600}$ = 1.2 in 12 l of YEP supplemented with 2% glycerol, then induced, harvested, lysed, and eluted off a calmodulin-affinity column as described above. Fractions containing ORC were pooled and applied to 1 ml BV Anti-M2 Flag resin (Sigma-Aldrich) which was pre-equilibrated with buffer A containing 0.2 M KCl and 0.02% NP-40. The Flag resin was incubated with rotation for 2 hr at 4°C after which it was collected on a column and the flow-through was discarded. The resin was washed with 10 BV of buffer A containing 0.2 M KCl and 0.02% NP-40. ORC was eluted with 5 BV buffer A containing 0.2 M KCl, 0.02% NP-40, and 0.3 mg/ml 3xFlag peptide with 30 min incubations before each elution. Fractions containing ORC were purified on SP Sepharose Fast Flow, sortase coupled to C-peptide labeled with DY-549P1, and further purified by gel filtration and Ni-NTA columns as described above. Peak fractions were aliquoted and stored at −80°C.

## Preparation of labeled ORC$^{1N-550}$

The *S. cerevisiae* strain yAZ55 ($^{Ubiquitin-GGG-3xFlag}$Orc1, Orc2-6) was grown to $OD_{600}$ = 1.2 in 8 l of YEP supplemented with 2% glycerol, then cell-cycle arrested, induced, harvested, and lysate prepared as described above. The lysate was applied to 1 ml BV Anti-M2 Flag resin (Sigma-Aldrich) which was pre-equilibrated with ORC lysis buffer. Flag resin was incubated with rotation for 2 hr at 4°C after which it was collected on a column and the flow-through was discarded. The resin was washed with 10 BV of buffer A containing 0.2 M KCl and 0.02% NP-40. ORC was eluted with 5 BV buffer A containing 0.2 M KCl, 0.02% NP-40, and 0.3 mg/ml 3xFlag peptide with 30-min incubations before each elution. Fractions containing ORC were pooled and purified on SP Sepharose Fast Flow (Cytiva) as described above. Fractions containing ORC were sortase coupled to N-peptide labeled with Dylight-550, and further purified by gel filtration and Ni-NTA columns as described above. Peak fractions were aliquoted and stored at −80°C.

## Preparation of labeled Mcm2-7$^{2C-649}$ and Mcm2-7$^{4N-650}$

The *S. cerevisiae* strains yST210 (Mcm2$^{C-LPETGG}$, 3xFlag-Mcm3, Mcm4-7, Cdt1) and yST180 (Mcm2, 3xFlag-Mcm3, $^{Ubiquitin-GGG}$Mcm4, Mcm5-7, Cdt1) were purified as described previously (*Ticau et al., 2015*; *Ticau et al., 2017*) with the following modifications. We used sortase to couple Mcm2$^{C-LPETGG}$ (yST210) to C-peptide conjugated to DY-649P1 and $^{GGG}$Mcm4 (yST180) to N-peptide conjugated to Dylight550. Following elution off the Anti-M2 Flag resin, peptide coupling and gel filtration (Superdex 200 10/300), peak fractions containing peptide-coupled Mcm2-7 were pooled. These pooled fractions were applied to 0.3 ml of Ni-NTA Agarose Resin (Qiagen) pre-equilibrated in buffer B with 10 mM imidazole. The resin was incubated for 1.5 hr with rotation at 4°C. The flow-through was discarded and the resin was washed sequentially with 3 ml buffer B with 25 mM imidazole, 50 mM imidazole, and 75 mM imidazole. Peptide-coupled Mcm2-7 was eluted using buffer B with 0.3 M imidazole. Peak fractions were aliquoted, and stored at −80°C.

## Preparation of labeled Mcm2-7$^{2C-650}$ and Mcm2-7$^{2C-650,5RA}$

The *S. cerevisiae* strains for Mcm2-7 expression in the absence of Cdt1, ySG12 (Mcm2$^{C-LPETGG}$), and ySG32 (Mcm2$^{C-LPETGG}$, Mcm5RA) were purified in the same manner as the Cdt1/Mcm2-7 complex as described previously (*Ticau et al., 2015*; *Ticau et al., 2017*). We used sortase to couple Mcm2$^{C-LPETGG}$ (ySG12 and ySG32) to C-peptide conjugated to Dylight650. The Ni-NTA selection for labeled protein was modified as described above for Mcm2-7$^{2C-649}$.

## Preparation of labeled Mcm2-7$^{3N-650}$ and Mcm2-7$^{3N-650,5RA}$

The *S. cerevisiae* strains for Mcm2-7 expression in the absence of Cdt1, ySG24 ($^{3xFlag-TEV[ENLYFQ/G]GG}$Mcm3), and ySG33 ($^{3xFlag-TEV[ENLYFQ/G]GG}$Mcm3, Mcm5RA) were grown, harvested, and lysate prepared in the same manner as the Cdt1–Mcm2-7 complex as described previously (*Ticau et al., 2015*). The lysate was applied to 1 ml BV Anti-M2 Flag resin (Sigma-Aldrich) in buffer C (50 mM HEPES–KOH pH 7.6, 5 mM Mg(OAc)$_2$, 10% glycerol, 0.3 M KGlu, 0.02% NP-40, 1 mM ATP). The Flag resin was incubated with rotation for 2 hr at 4°C after which it was collected on a column and the flow-through was discarded. The resin was washed with buffer C and resuspended to create a 50% slurry. $^{GGG}$Mcm3 (in association with the other Mcm2-7 subunits) was cleaved off the resin with 500 units of 7× His TEV protease (NEB; rotation overnight at 4°C). The flow-through was collected and applied to 0.4 ml volume Ni-NTA Agarose resin (Qiagen) to remove the 7× His TEV protease. The flow-through containing cleaved Mcm2-7 (with the three N-terminal glycines revealed) was coupled to N-peptide labeled with Dylight-650 using sortase, and further purified by gel filtration and Ni-NTA columns as described above. Peak fractions were aliquoted and stored at −80°C.

## Preparation of labeled Cdt1$^{N-649}$ and Cdt1$^{C-650}$

The *S. cerevisiae* strains for Cdt1 purification yST103 ($^{Ubiquitin-GGG}$Cdt1-Flag) and ySG046 (3xFlag-Cdt1$^{C-LPETGG}$) were grown to OD$_{600}$ = 1.2 in 8 l of YEP supplemented with 2% glycerol, then cell-cycle arrested, induced, harvested, and lysate prepared as described for the Cdt1/Mcm2-7 complex (*Ticau et al., 2015*). The lysate was applied to 1 ml BV Anti-M2 Flag resin (Sigma-Aldrich) in buffer D (50 mM HEPES–KOH pH 7.6, 5 mM Mg(OAc)$_2$, 10% glycerol, 150 mM KGlu, 1 mM ATP, 0.02% NP-40) and eluted with buffer D containing 0.3 mg/ml 3xFlag peptide. Fractions containing Cdt1 were sortase coupled to N-peptide labeled with DY649-P1 (for Cdt1$^{N-649}$) or C-peptide labeled with Dylight650 (for Cdt1$^{C-650}$) as described for ORC above. The reaction was applied to a Superdex 75 10/300 gel filtration column equilibrated in buffer D containing 10 mM imidazole. Peak fractions containing peptide-coupled Cdt1 were pooled and incubated with 0.3 ml of Ni-NTA Agarose Resin (Qiagen) pre-equilibrated in buffer D with 10 mM imidazole for 1.5 hr with rotation at 4°C. The flow-through was discarded and the resin was washed sequentially with 3 ml buffer D with 25, 50, and 75 mM imidazole. Peptide-coupled Cdt1 was eluted using buffer D with 0.3 M imidazole. Peak fractions were pooled, aliquoted, and stored at −80°C.

## Single-molecule assay for helicase loading

The micromirror total internal reflection microscope used for single-molecule helicase-loading experiments (excitation wavelengths 488, 532, and 633 nm) is described in *Friedman and Gelles, 2012*; *Friedman et al., 2006*. Single-molecule reactions were performed as described in *Ticau et al., 2015* with the following modifications. The surface of the reaction chamber was cleaned by sonication with 2% Micro-90 (Cole Parmer), 0.1 M KOH, and 100% ethanol sequentially for 1 hr each. The cleaned chamber was then derivatized with a 500:1 ratio of mPEG-silane-2000 (creative PEGWorks) and biotin-mPEG-silane-3400 (Laysan Bio). Streptavidin-labeled fiducial markers (0.04 µm, Invitrogen TransFluoSpheres) were flowed into the reaction chamber at a 1:400,000 dilution. A biotinylated 1.3-kb-long origin DNA template (*ARS1*) labeled with Alexa Fluor 488 was coupled to the surface of a reaction chamber using streptavidin. DNA molecules were identified by acquiring two to three images with 488 nm excitation at the start of the experiment. All helicase-loading reactions and imaging were performed in the reaction buffer as described previously (*Kang et al., 2014*; *Ticau et al., 2015*) containing 25 mM HEPES–KOH pH 7.6, 7.5 mM Mg(OAc)$_2$, 50 µM Zn(OAc)$_2$, 100 µM ethylenediaminetetraacetic acid (EDTA), 3 mM ATP, 0.3 M KGlu, 12.5 mM dithiothreitol (DTT), 2.5 mg/ml bovine serum albumin (BSA), an oxygen scavenging system (glucose oxidase/catalase), and triplet state quenchers to minimize photobleaching (*Friedman et al., 2006*; *Hoskins et al., 2011*). All reactions

additionally contained 1 μM of a 60 bp double-stranded DNA that cannot compete for ORC binding in a sequence-specific manner (*Bell and Stillman, 1992*). Concentrations of the helicase-loading proteins used were 10 nM Cdc6, 0.5 nM ORC, 10–15 nM Mcm2-7, and 10–20 nM Cdt1, with the exception of reactions in *Figure 3* (see below). After helicase-loading proteins were perfused into the DNA-decorated chamber, 1-s duration images were acquired by alternate excitation with 532 and 633 nm lasers (laser powers were 0.7 and 0.35 mW, respectively, as measured incident to the micromirror). The average dead time between consecutive images was 0.2 s. The 2.4-s cycle was repeated 600 times in the course of an experiment (~24 min). Emission was observed using dual-view optics that separated emission wavelengths <635 nm (blue- and green-excited dyes) and >635 nm (red-excited dyes).

## Single-molecule helicase-loading assays mixtures of two labeled proteins

The helicase-loading reactions described in *Figures 2 and 3* were performed with mixtures of N- and C-terminally labeled preparations of Mcm2-7 or ORC, respectively. In experiments with mixed Mcm2-7 preparations (*Figure 2*), 7.5 nM Cdt1/Mcm2-7$^{2C-649}$ and 7.5 nM Cdt1/Mcm2-7$^{4N-650}$ were mixed and added to a reaction with 0.5 nM ORC and 10 nM Cdc6. In experiments with mixed ORC preparations, 0.25 nM ORC$^{1N-550}$ was mixed with 0.25 nM ORC$^{5C-549}$. Standard concentrations of Cdt1/Mcm2-7$^{2C-649}$ (15 nM) and Cdc6 (10 nM) were used.

## Staged single-molecule helicase-loading assay

The helicase-loading reaction with prebound ORC/Cdc6 described in *Figure 5e, f* was performed in a staged manner. 1 nM ORC$^{6C-549}$ and 10 nM Cdc6 were incubated with slide-attached *ARS1* origin DNA in the helicase-loading reaction buffer described above. After 10 min, unbound proteins were washed away with a low-salt wash buffer containing 50 mM HEPES–KOH pH 7.6, 5 mM Mg(OAc)$_2$, and 300 mM KGlu. The helicase-loading reaction was performed by adding all proteins (except ORC) at the following concentrations: Mcm2-7$^{3N-650}$ (15 nM), Cdt1 (20 nM), and Cdc6 (10 nM).

## FRET data analysis

CoSMoS datasets were analyzed as described previously (*Ticau et al., 2015*), except that records of protein fluorescence were corrected for background fluorescence as described in *De Jesús-Kim et al., 2021*. Records of acceptor-excited fluorescence in *Figure 2* were additionally corrected for spatial variations in local laser excitation as described previously (*De Jesús-Kim et al., 2021*). Donor-excited records ($D_{ex}$, $D_{em}$ and $D_{ex}$, $A_{em}$) used to calculate $E_{FRET}$ were not subject to the corrections for local laser excitation in any instance.

By alternating between laser excitation wavelengths, we monitored the co-localization of both the donor fluorophore on ORC and acceptor fluorophore(s) on Mcm2-7 and/or Cdt1 with origin DNA molecules. Double-hexamer formation events with no labeled ORC (donor) were excluded from data analysis. To determine the time of formation of the high-FRET state ($E_{FRET}$ 0.7–0.8), we noted the earliest time the >635 nm emission FRET signal rose above background noise level while a donor fluorophore was present (as determined by monitoring the signal in the >635 nm field when only the 532 nm laser was turned on). Only spots where the arrival of the acceptor fluorophore could be confirmed within 4.8 s were considered genuine instances of high FRET.

Apparent FRET efficiency was calculated using:

$$E_{FRET} = I_{Acceptor}/(I_{Acceptor} + I_{Donor})$$

where $I_{Acceptor}$ and $I_{Donor}$ are the background-corrected acceptor and donor emission intensities, respectively, that are observed during donor excitation (*De Jesús-Kim et al., 2021*). Because the intensities are background corrected, fluctuations in $I_{Acceptor}$ and $I_{Donor}$ sometimes result in $E_{FRET}$ values below 0 or above 1. Rare $E_{FRET}$ values greater than +2 or less than −2 were left out of the $E_{FRET}$ histograms. Confidence bounds for kinetic data plotted as cumulative distributions were determined using the Greenwood formula as implemented in the Matlab 'ecdf' function. We note that each single-molecule event observed is an independent, separately observed replicate of the helicase-loading reaction. Nevertheless, the data in most figures in the main text come from combining two to five instances of an experiment, each being the result of a separate helicase-loading reaction. The only

exception is *Figure 5e,f*, where the ORC$^{6\Delta N119}$ mutant and prebound ORC$^{6C-549}$/Cdc6 data each come from a single experimental instance.

## Determining ORC$^{5C-549}$ labeling fraction

The fraction of ORC$^{5C-549}$ molecules that were fluorescently labeled was determined as described previously (*Ticau et al., 2015*). Briefly, 10 µl (~5.6 µg) of labeled ORC$^{5C-549}$ was mixed with maleimide-DY-649P1 dissolved in anhydrous DMSO, in a 1:1 molar ratio at 4°C for 10 min. The reaction was terminated with 2 mM DTT. To ensure that we were monitoring fully assembled and functional ORC complexes, we added 1 nM of the double-labeled ORC to a single-molecule reaction chamber containing origin DNA and monitored ORC-DNA colocalization. The fraction of maleimide-DY-649P1-labeled ORC molecules that also contained DY-549P1 was determined and reported as the percent labeling by the DY-549P1.

## Determining ORC$^{5C-549}$ photobleaching rate

To determine the photobleaching rate of ORC$^{5C-549}$, single-molecule helicase-loading reactions were performed with 0.5 nM ORC$^{5C-549}$, 15 nM unlabeled Cdt1-Mcm2-7$^{5RA}$, and 10 nM Cdc6 under two laser exposure conditions (*Figure 1—figure supplement 4*). Data for the experiments shown throughout the manuscript was acquired at a relative laser exposure of 1, by alternating between the two laser excitation wavelengths. For the ORC$^{5C-549}$ photobleaching experiments, relative exposure of 1 was achieved by alternating acquisition of 1-s frames with excitation with the 532 nm laser and 1-s frames with no excitation (dead time between frames, 0.2 s). The relative laser exposure was increased to 2.4 by continuously acquiring 1-s frames with 532 nm excitation and no dead time. ORC dwell times under the two laser exposures were each fit to single-exponential functions using a maximum likelihood algorithm and bootstrap methods were used to determine uncertainty estimates, similar to what was previously described (*Crawford et al., 2013*; *Friedman and Gelles, 2012*).

## Acknowledgements

We are grateful to members of the Bell Laboratory for useful discussions. We thank Xiaoxue (Snow) Zhou and Hazal B Kose for comments on the manuscript, Alexandra M Pike and Paritosh Gangar-amani for comments on the figures, and Christian Ramsoomair for preparation of a subset of proteins described in this manuscript. This work was supported by NIH grants GM52339 (SPB) and R01 GM81648 (JG). SG was supported in part by an NIH Pre-Doctoral Training Grant (T32 GM007287) and a MathWorks Science Fellowship. SPB is an investigator with the Howard Hughes Medical Institute. This work was supported in part by the Koch Institute Support Grant P30-CA14051 from the NCI. We thank the Biopolymers core of the Koch Institute Swanson Biotechnology Center for technical support.

## Additional information

### Funding

| Funder | Grant reference number | Author |
|---|---|---|
| Howard Hughes Medical Institute | Investigator Award | Stephen P Bell |
| National Institute of General Medical Sciences | GM52339 | Stephen P Bell |
| National Institute of General Medical Sciences | GM81648 | Jeff Gelles |
| National Institute of General Medical Sciences | GM007287 | Shalini Gupta |
| National Cancer Institute | P30-CA14051 | Stephen P Bell |

The funders had no role in study design, data collection, and interpretation, or the decision to submit the work for publication.

## Author contributions
Shalini Gupta, Data curation, Formal analysis, Investigation, Methodology, Validation, Visualization, Writing - original draft, Writing - review and editing; Larry J Friedman, Formal analysis, Methodology, Resources, Software, Supervision, Writing - review and editing; Jeff Gelles, Formal analysis, Funding acquisition, Project administration, Supervision, Writing - review and editing; Stephen P Bell, Conceptualization, Funding acquisition, Methodology, Project administration, Supervision, Writing - original draft, Writing - review and editing

## Author ORCIDs
Shalini Gupta ⓘ http://orcid.org/0000-0002-5446-7912
Larry J Friedman ⓘ http://orcid.org/0000-0003-4946-8731
Jeff Gelles ⓘ http://orcid.org/0000-0001-7910-3421
Stephen P Bell ⓘ http://orcid.org/0000-0002-2876-610X

## Decision letter and Author response
Decision letter https://doi.org/10.7554/eLife.74282.sa1
Author response https://doi.org/10.7554/eLife.74282.sa2

---

# Additional files

## Supplementary files
• Transparent reporting form

## Data availability
Source data for the single-molecule experiments is provided as "intervals" files that can be read and manipulated by the Matlab program imscroll, which is publicly available: https://github.com/gelles-brandeis/CoSMoS_Analysis (copy archived at swh:1:rev:3eec2cbfa54018389fc1905b54c-4b062723a5a7f). These data can also be read directly in Matlab. The source data are archived at: https://doi.org/10.5061/dryad.547d7wm8z.

The following dataset was generated:

| Author(s) | Year | Dataset title | Dataset URL | Database and Identifier |
|---|---|---|---|---|
| Gupta S | 2021 | A Helicase-tethered ORC Flip Enables Bidirectional Helicase Loading | https://doi.org/10.5061/dryad.547d7wm8z | Dryad Digital Repository, 10.5061/dryad.547d7wm8z |

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
