## [Editor Report]

The initiation of DNA replication in eukaryotes is preceded by the assembly of a pre-Replicative Complex (pre-RC) at all potential origins of DNA replication during the G1 phase of the cell division cycle. The pre-RC contains a double hexamer of Mcm2-7 subunits and each hexamer eventually becomes the core of the two replicative helicases during the initiation of DNA synthesis. The current paper addresses the role of ORC and Cdc6 in loading the Cdt1-bound Mcm2-7 hexamer onto origin DNA and the data show that a single ORC molecule can load two Mcm2-7 hexamers in a sequential loading reaction that involves ORC flipping on the origin DNA. The results nicely complement other studies that show a detailed pathway for pre-RC assembly.

---

## [Decision Letter]

**Decision letter after peer review:**

Thank you for submitting your article "A Helicase-tethered ORC Flip Enables Bidirectional Helicase Loading" for consideration by *eLife*. Your article has been reviewed by 3 peer reviewers, including Bruce Stillman as Reviewing Editor and Reviewer #1, and the evaluation has been overseen by Kevin Struhl as the Senior Editor.

The reviewers have discussed their reviews with one another, and the Reviewing Editor has drafted this to help you prepare a revised submission. Essentially there are some suggestions that request clarifications of the data and additions to the manuscript to make it more clear.

Essential revisions:

These comments are mostly for clarification and discussion

1) While the abstract correctly states that a single ORC can load a double hexamer of Mcm2-7, their own data shows that at a low frequency, two ORC molecules can do the reaction. This should be so stated.

2) The abstract says "quick succession" – It might be more forceful to say "within 8 seconds", considering that this step truly appears quick, compared to the first loading, and possibly the second.

3) In the early part of the Intro – it would be helpful to mention that yeast origins are heterogeneous in sequence, with ORC binding elements that have various distances from one another. Many readers could assume that each origin is identical in this respect, given that many bacterial origins, plasmid origins, and eukaryotic viral origins have very defined sequence and spacer requirements. True – this aspect is addressed in the Discussion to some extent – but the naïve reader should be more informed early on in the Intro.

4) In the 3rd paragraph of the intro – the Costa EM work (Miller 2019) is cited. It would be good to identify the origin used for that study.

5) Figure 2 results. The reader might have a hard time to understand that one is mixing all these proteins, and that all possible outcomes are being shown. This brings up the question of whether the reaction could be staged, for an easier interpretation of the entire paper (i.e. first bind ORC-Cdc6, or OCCM-ATPgS, and then proceed in the presence of excess unlabeled ORC). It is one reviewer's impression from E. Greene's work (maybe others?) that the ORC-Cdc6 is stably bound to ARS, and so is OCCM. So – it seems that this strategy could have been used. Is there a reason, for example regarding the instrumentation, that this is not the case? This question will come up in many readers minds, so it might as well be addressed early on.

6) On the bottom of page 6, the authors describe the SM events that occur. 28% of the molecules showed a two-step loading of Mcm2-7, and only 12% were analyzed because of the duration of the Mcm2-7 association. What are the remaining 88% of reactions? A comment on this is appropriate.

Of the 28% of DNA molecules with two increases in Mcm2-7 fluorescence, only 12% were long-lived and included in the data analysis (page 6). It is not entirely clear why the other 16% were excluded. Do they correspond to abortive loading events? A brief explanation in the main text would be helpful.

The Figure 1 results indicate 96 observables, which is fine – but is this all on one TIRF chip set-up, or does this number of observables require the sum of many different chip data sets? This same Q applies throughout. In this reviewer's mind, it the 96 observables are obtained for one TIRF chip set-up – the experiment should be repeated at least 3 times for a statistical reproducibility check.

7) Figure 1: The FRET results are shown in blue, but in fact it is red fluorophore emission that is excited by the green fluorophore excitation (correct?). In which case, it is a just a little bit confusing to see a blue line plot. It seems it should be red. This is the case in other Figures too. However, I like the blue. But – it should be stated in the legend that the blue line is the observed fluorescence for excitation at the green fluorophore, and the emission wavelength of the red fluorophore.

8) The inability of ORC-Orc6deltaN119 to form MO interactions would predict that the average dwell time of this mutant ORC on DNA should be shorter than that of WT ORC since it will not flip and rebind DNA after loading of the first MCM. Have the authors looked whether the data is consistent with this prediction? This could be added to Figure 5 – figure supplement 3.

9) On page 8 (second paragraph), the authors refer to three types of two-ORC double hexamer formation events, but only two of them, 8/166 and 14/166, are described. What is the third type?

10) In Figure 1c, can the authors indicate in the total emission trace the areas corresponding to protein-induced fluorescence enhancement? Given the noise in the traces, this enhancement is not easily recognizable to non-experts.

11) On page 9 (4th line from bottom) and in the legend of Figure 1-figure supplement 3, should 'Rem' be 'Aem'?

12) On page 18, the authors report that there is strong FRET between Cdt1 N-649 and ORC 5C-549, while Figure 4-figure supplement 1 states and shows the opposite. Can the authors clarify this inconsistency?

13) Figure 6 – figure supplement 3a should probably refer to Figure 6b instead of Figure 6c. In the legend of Figure 6 – figure supplement 3b, it seems that Mcm2-7 2C-650 should be Mcm2-7 3N-650.

14) Diffley/Dekker single molecule experiments, published recently (this year) indicate that they see single hexamers of MCM loading. Do the authors of the current study ever see single MCM loading?

15) Where are the ATP steps? Have the authors examined this? There are so many ATP requiring proteins in this system, and it would be good to have the discussion summarize the one, or more ATP sites, in which subunits, are needed for the steps in Figure 7. Alternatively, for the authors to summarize what is not yet known about ATP site identity or stoichiometry.

16) Figure 7. This is a very detailed model and will likely be reproduced in reviews. I suggest that the authors include ORC-Cdc6 induced DNA bending in the model because this is critical for the loading reaction as demonstrated by Feng et. al 2021. Indeed, DNA bending could place the second Mcm2-7 molecule in the correct orientation for the double hexamer formation.

17) Doesn't ORC bind throughout the cell cycle? Yet, it is indicated that ORC is ejected in the process of double hexamer formation. If possible, it would be nice to shed more light on this (i.e. by discussion, not experimentation).

18) They find that OM rapidly moves to a MO that anticipates a second M. This is supported by an ORC mutant? I find this pretty amazing. They presume O uses the first M as a tether while searching for a second O site. Do they see if one MCM is loaded – ever? (i.e. this overlaps with # 13).

118 Under the title "Comparison of one-ORC and two-ORC mechanisms for helicase loading"" the concentration of ORC in the cell is touched upon. It would be good to let readers know that replication occurs in nuclear foci, and that intracellular concentration, which is dependent on proximity effects, cannot be known by simply determining the cellular abundance of a protein.

Two comments from the reviewers suggested additional experiments and while the consensus was that these experiments are not essential, nevertheless they are reported below to the authors so that they can consider them.

1) The authors propose that the N-terminal domain of Orc6 establishes a tether with Mcm2-7 to prevent ORC release into solution before MO and B2 interactions are formed but were not able to investigate the timing of Orc6N binding because N-terminal labeling of Orc6 interfered with protein function. Can the authors use an internal label to establish the timing of these events? It seems that forming such a tether prior to ORC release from the original DNA binding site would be key to prevent ORC dissociation into solution before MO interactions can occur.

2) Related to the previous point, the linker between Orc6 NTD and CTD is much shorter in higher eukaryotes, which would make it unlikely that Orc6N could tether ORC to the first MCM during the flip and prior to MO formation in these systems. Have the authors tried reducing Orc6 linker length? Does this still support Mcm2-7 loading by the proposed one-ORC-flip mechanism or does it shift loading towards a two-ORC mechanism?

---

## [Author Response]

Essential revisions:These comments are mostly for clarification and discussion1) While the abstract correctly states that a single ORC can load a double hexamer of Mcm2-7, their own data shows that at a low frequency, two ORC molecules can do the reaction. This should be so stated.

We have changed the abstract to include the statement that “*In the large majority of events*, we observed a single ORC molecule recruiting both Mcm2-7-Cdt1 complexes…”

2) The abstract says "quick succession" – It might be more forceful to say "within 8 seconds", considering that this step truly appears quick, compared to the first loading, and possibly the second.

We have modified the abstract to include the fact that the time frame is within seconds, but prefer not to emphasize the 8 s timing because it is an average of a distribution of values from a number of single-molecule observations. In fact, the large majority of MO formation events fall within the 3.6 ± 1.2 s bin after first Cdt1 release in Figure 6c. We have made the following change to the abstract:

“Between first and second helicase recruitment, a rapid change in interactions between ORC and the first Mcm2-7 occurs. *Within seconds*, ORC breaks the interactions mediating first Mcm2-7 recruitment…”

3) In the early part of the Intro – it would be helpful to mention that yeast origins are heterogeneous in sequence, with ORC binding elements that have various distances from one another. Many readers could assume that each origin is identical in this respect, given that many bacterial origins, plasmid origins, and eukaryotic viral origins have very defined sequence and spacer requirements. True – this aspect is addressed in the Discussion to some extent – but the naïve reader should be more informed early on in the Intro.

We have modified paragraph 3 of the introduction to include a description of origin architecture in *S. cerevisiae*:

“Multiple mechanisms have been proposed to explain how two oppositely-oriented helicases are loaded at an origin. In addition to a primary ORC-binding site, natural origins include at least one additional weaker, inverted ORC-binding site (Chang et al., 2011; Palzkill and Newlon, 1988; Wilmes and Bell, 2002). These sequences can be located at a variety of distances from one another but are typically less than ~ 60 bp apart (Chang et al., 2011)”

4) In the 3rd paragraph of the intro – the Costa EM work (Miller 2019) is cited. It would be good to identify the origin used for that study.

Miller et al., 2019 used the *ARS1* origin, which is the same origin that previous single-molecule studies have used (Ticau et al., 2015, 2017). We have incorporated this information in the introduction:

“In contrast, single-molecule helicase loading experiments with the *ARS1* origin showed that a single ORC molecule can direct loading of both Mcm2 7 helicases (Ticau et al., 2015). A single-ORC model is also supported by time-resolved cryoelectron microscopy (cryo-EM) experiments showing predominantly one ORC molecule bound to the DNA in each helicase-loading intermediate observed on the *ARS1* origin (Miller et al., 2019). A goal of the current studies is to address these apparently contradictory observations.”

5) Figure 2 results. The reader might have a hard time to understand that one is mixing all these proteins, and that all possible outcomes are being shown. This brings up the question of whether the reaction could be staged, for an easier interpretation of the entire paper (i.e. first bind ORC-Cdc6, or OCCM-ATPgS, and then proceed in the presence of excess unlabeled ORC). It is one reviewer's impression from E. Greene's work (maybe others?) that the ORC-Cdc6 is stably bound to ARS, and so is OCCM. So – it seems that this strategy could have been used. Is there a reason, for example regarding the instrumentation, that this is not the case? This question will come up in many readers minds, so it might as well be addressed early on.

We thank the reviewer for this suggestion that would further test whether one ORC can perform helicase loading. We performed a staged experiment in which we pre-bound ORC/Cdc6 to *ARS1* DNA, washed away all unbound proteins, and then added all the helicase loading proteins (Mcm2-7, Cdt1 and Cdc6) except ORC. Although the number of Mcm2-7 binding events observed under these conditions was reduced (likely due to ORC/Cdc6 dissociation from DNA during the wash step), we observed clear instances of double-hexamer formation. Importantly, in all double hexamer formation events with pre-bound ORC/Cdc6, MO formation was observed and it anticipated recruitment of the second Mcm2-7. Additionally, the fraction of first Mcm2-7 molecules with MO-FRET establishment and double hexamer formation was similar to that observed for reactions in which ORC was continuously present. We have added these results to the paper (see new section on page 25 “Pre-bound ORC/Cdc6 can form double hexamers without additional ORC in solution”, and modified Figure 5e and 5f). These results are as predicted for helicase loading mediated by a single ORC DNA-binding event.

6) On the bottom of page 6, the authors describe the SM events that occur. 28% of the molecules showed a two-step loading of Mcm2-7, and only 12% were analyzed because of the duration of the Mcm2-7 association. What are the remaining 88% of reactions? A comment on this is appropriate.Of the 28% of DNA molecules with two increases in Mcm2-7 fluorescence, only 12% were long-lived and included in the data analysis (page 6). It is not entirely clear why the other 16% were excluded. Do they correspond to abortive loading events? A brief explanation in the main text would be helpful.

As this reviewer points out, 28% of the total DNA molecules have two sequential increases in Mcm2-7 fluorescence. 12% of these events have long-lived second Mcm2‑7 associations ( >= 20 frames or 48 seconds). The remaining 16% events are instances in which the second Mcm2-7 is associated for a short period of time (<20 frames of acquisition). Since we observed OM-FRET throughout the second Mcm2-7 association in most of the short events, we conclude that these are unsuccessful helicase-loading events where the second Mcm2-7 is recruited but fails to remain associated with the origin. We have added the following clarification (italic portion) to the main text (bottom of page 6):

“These long-lived sequential associations occurred on 12% of DNAs and represent successful Mcm2-7 double-hexamer formation (Ticau et al., 2015). The remaining sequential associations (on 16% of DNAs) were short-lived (<48 s), and we considered these to be unsuccessful instances of helicase loading (Ticau et al., 2015).”

The Figure 1 results indicate 96 observables, which is fine – but is this all on one TIRF chip set-up, or does this number of observables require the sum of many different chip data sets? This same Q applies throughout. In this reviewer's mind, it the 96 observables are obtained for one TIRF chip set-up – the experiment should be repeated at least 3 times for a statistical reproducibility check.

We note that each single-molecule double-hexamer formation event observed is an independent, separately observed replicate of the helicase-loading reaction. For example, in Figure 1 we have 106 such replicates of the reaction. The data in that and most figures in the manuscript comes from combining data from multiple instances of an experiment, each being the result of a separate helicase-loading reaction (Author response Table 1). The only exceptions are the data for the ORC^6∆N119^ mutant in Figure 5e-f where we had 418 Mcm2-7 associations from a single replicate, and the experiment we added in response to reviewer comment 5 (pre-bound ORC; Figure 5e-f). The multiple instances are used simply to increase the number of molecules observed in the experiment.

**Author response table 1. sa2table1:** Figure numbers from main text and the corresponding number of replicates.

Figure number	Number of experimental instances
Figure 1 and Figure 4a	4
Figure 2	2
Figure 3	4
Figure 4d	3
Figure 5c-f (wild-type)	3
Figure 6	5

For example, the data in Figure 3d is combined from four experimental instances. Double-hexamer formation events from each are categorized based on OM-FRET profile as CC, NN, CN and NC (Author response Table 2). Each of the instances individually strongly supports the conclusion that the majority of events are of the CC and NN types. There is no indication of any significant difference in the behavior of molecules from one instance to another.

**Author response table 2. sa2table2:** Fraction of events (± S.E). for each instance broken down by the associated FRET profile.

	CC	NN	CN	NC
Fig. 3 Instance 1 (N = 17)	0.47 ± 0.12	0.47 ± 0.12	0.00 ± 0.00	0.06 ± 0.06
Fig. 3 Instance 2 (N = 22)	0.41 ± 0.10	0.59 ± 0.10	0.00 ± 0.00	0.00 ± 0.00
Fig. 3 Instance 3 (N = 25)	0.48 ± 0.10	0.48 ± 0.10	0.04 ± 0.04	0.00 ± 0.00
Fig. 3 Instance 4 (N = 18)	0.44 ± 0.12	0.44 ± 0.12	0.00 ± 0.00	0.11 ± 0.07
Total (N = 82)	0.45 ± 0.05	0.50 ± 0.06	0.01 ± 0.01	0.04 ± 0.02

We have also added a version of this response to the methods:

“We note that each single-molecule event observed is an independent, separately observed replicate of the helicase-loading reaction. Nevertheless, the data in most figures in the main text comes from combining 2-5 instances of an experiment, each being the result of a separate helicase-loading reaction. The only exception is Figure 5e-f, where the ORC^6∆N119^ mutant and pre-bound ORC^6C‑549^/Cdc6 data each come from a single experimental replicate.”

7) Figure 1: The FRET results are shown in blue, but in fact it is red fluorophore emission that is excited by the green fluorophore excitation (correct?). In which case, it is a just a little bit confusing to see a blue line plot. It seems it should be red. This is the case in other Figures too. However, I like the blue. But – it should be stated in the legend that the blue line is the observed fluorescence for excitation at the green fluorophore, and the emission wavelength of the red fluorophore.

The blue graphs not green-excited red emission as suggested by the reviewer. They are apparent FRET efficiency (*E*_FRET_), which is the ratio of green-excited red emission to green-excited total (i.e., red plus green) emission. We explain in the Figure 1c caption “*E*_FRET_ values are calculated using donor-excited emission from the donor and acceptor fluorophores (see Methods)” and the caption and the figure both explicitly indicate that the blue record is “*E*_FRET_” corresponding to “D_ex_, [A_em_/ (A_em_ + D_em_)]” Given that *E*_FRET_ is derived from the measurement of *both* green and red emission, we feel that using a third color (e.g., blue) is more appropriate than showing *E*_FRET_ records as red. Emission wavelengths are given in the Methods.

8) The inability of ORC-Orc6deltaN119 to form MO interactions would predict that the average dwell time of this mutant ORC on DNA should be shorter than that of WT ORC since it will not flip and rebind DNA after loading of the first MCM. Have the authors looked whether the data is consistent with this prediction? This could be added to Figure 5 – figure supplement 3.

As the reviewer suggests, this is indeed the case. We have added two panels to Figure 5- figure supplement 3 (e and f) to describe how ORC dwell times differ between the ORC^6∆N119^ mutant and wild-type ORC. We show that the ORC^6∆N119^ mutant has shorter ORC dwell times on DNA compared to wild-type ORC. Further, in the wild-type ORC dataset, we find that establishing MO interactions is what extends ORC dwell times during helicase loading. WT ORC molecules that do not form the MO complex have a similar distribution of dwell times as ORC^6∆N119^ (compare panels a and c in Figure 5 – figure supplement 3).

9) On page 8 (second paragraph), the authors refer to three types of two-ORC double hexamer formation events, but only two of them, 8/166 and 14/166, are described. What is the third type?

There are only two types of two-ORC events described. We have corrected this error in the text.

10) In Figure 1c, can the authors indicate in the total emission trace the areas corresponding to protein-induced fluorescence enhancement? Given the noise in the traces, this enhancement is not easily recognizable to non-experts.

We have changed Figure 1—figure supplement 3 to use dashed lines to more clearly identify the part of the emission trace in which we infer protein-induced fluorescence enhancement (PIFE) is observed.

11) On page 9 (4th line from bottom) and in the legend of Figure 1-figure supplement 3, should 'Rem' be 'Aem'?

We thank the reviewer for their close reading of this manuscript and for pointing out this error. We have corrected it in the text.

12) On page 18, the authors report that there is strong FRET between Cdt1 N-649 and ORC 5C-549, while Figure 4-figure supplement 1 states and shows the opposite. Can the authors clarify this inconsistency?

The average *E*_FRET_ observed when Cdt1 is present was 0.358 ± 0.003, which is easily distinguished from that of the OM-FRET pair (average *E*_FRET_ 0.72 ± 0.01). The text should say “Although Cdt1^N‑649^ is also labeled with an acceptor fluorophore, this site of labeling does *not* exhibit strong FRET with ORC^5C‑549”^. We have made this correction to the text.

13) Figure 6 – figure supplement 3a should probably refer to Figure 6b instead of Figure 6c. In the legend of Figure 6 – figure supplement 3b, it seems that Mcm2-7 2C-650 should be Mcm2-7 3N-650.

Yes, we have corrected these errors in the text and legends.

14) Diffley/Dekker single molecule experiments, published recently (this year) indicate that they see single hexamers of MCM loading. Do the authors of the current study ever see single MCM loading?

As we reported in our original single-molecule studies of helicase loading (Ticau et al., 2015), we observe associations of single loaded Mcm2-7 molecules. In our experiments, only a fraction (0.18 ± 0.02) of first Mcm2‑7 binding events convert to stable second Mcm2‑7 recruitment. Although many of these Mcm2-7 molecules are released, a subset have longer dwell times and may represent loaded single hexamers (Figure 5—figure supplement 3c). We did not investigate these events further as the current studies were focused on the mechanism of double-hexamer formation.

15) -Where are the ATP steps? Have the authors examined this? There are so many ATP requiring proteins in this system, and it would be good to have the discussion summarize the one, or more ATP sites, in which subunits, are needed for the steps in Figure 7. Alternatively, for the authors to summarize what is not yet known about ATP site identity or stoichiometry.

We have added a paragraph to the discussion that addresses the potential roles for ATP hydrolysis during helicase loading (Page 32).

16) Figure 7. This is a very detailed model and will likely be reproduced in reviews. I suggest that the authors include ORC-Cdc6 induced DNA bending in the model because this is critical for the loading reaction as demonstrated by Feng et. al 2021. Indeed, DNA bending could place the second Mcm2-7 molecule in the correct orientation for the double hexamer formation.

We have made this change to the model in Figure 7 and Figure 7 —figure supplement 1.

17) Doesn't ORC bind throughout the cell cycle? Yet, it is indicated that ORC is ejected in the process of double hexamer formation. If possible, it would be nice to shed more light on this (i.e. by discussion, not experimentation).

Our previous single-molecule studies established that ORC is released with the second Cdt1 molecule during helicase loading (Ticau et al., 2015, 2017). However, once the double-hexamer has formed, it is certainly possible that ORC re-associates with DNA. The original ChIP studies of ORC showed reduced ORC binding during G1 (vs G2/M), suggesting that having loaded Mcm2-7 reduces ORC binding (Aparicio et al., 1997). We suggest that this is due to a combination of ORC release after loading and obscuring of the ACS by loaded Mcm2-7 after helicase loading. It is also worth noting that although ChIP detects ORC binding to origins throughout the cell cycle, this assay does not distinguish between long and short associations with the DNA. Although interesting, we do feel that adding this discussion would be distracting since it is fairly distant from the main points of the current paper.

18) They find that OM rapidly moves to a MO that anticipates a second M. This is supported by an ORC mutant? I find this pretty amazing. They presume O uses the first M as a tether while searching for a second O site. Do they see if one MCM is loaded – ever? (i.e. this overlaps with # 13).

Please see response to point 13 above.

118 Under the title "Comparison of one-ORC and two-ORC mechanisms for helicase loading"" the concentration of ORC in the cell is touched upon. It would be good to let readers know that replication occurs in nuclear foci, and that intracellular concentration, which is dependent on proximity effects, cannot be known by simply determining the cellular abundance of a protein.

We have added a discussion on liquid phase-separation recently reported for metazoan ORC (Parker et al., 2019; Hossain et al., 2021) on Page 34-35.

Two comments from the reviewers suggested additional experiments and while the consensus was that these experiments are not essential, nevertheless they are reported below to the authors so that they can consider them.1) The authors propose that the N-terminal domain of Orc6 establishes a tether with Mcm2-7 to prevent ORC release into solution before MO and B2 interactions are formed but were not able to investigate the timing of Orc6N binding because N-terminal labeling of Orc6 interfered with protein function. Can the authors use an internal label to establish the timing of these events? It seems that forming such a tether prior to ORC release from the original DNA binding site would be key to prevent ORC dissociation into solution before MO interactions can occur.

Although very interesting, this is a difficult experiment as we have yet to obtain a labeled form of the Orc6 N-terminal domain that is functional for helicase loading.

2) Related to the previous point, the linker between Orc6 NTD and CTD is much shorter in higher eukaryotes, which would make it unlikely that Orc6N could tether ORC to the first MCM during the flip and prior to MO formation in these systems. Have the authors tried reducing Orc6 linker length? Does this still support Mcm2-7 loading by the proposed one-ORC-flip mechanism or does it shift loading towards a two-ORC mechanism?

We thank the reviewer for this suggestion, and this is indeed a set of experiments that we would like to do in the future. However, we think that the addition of this data would not provide further evidence to support the conclusion we make in the current manuscript about whether one ORC can load both Mcm2-7 complexes in a double hexamer.